# IoDM: A Study on a IoT-Based Organizational Deception Modeling with Adaptive General-Sum Game Competition

Sang Seo [ID] and Dohoon Kim *[ID]

Department of Computer Science, Kyonggi University, Suwon-si 16227, Korea; tjtkd8271@kyonggi.ac.kr
* Correspondence: karmy01@kyonggi.ac.kr

**Abstract:** Moving target defense (MTD) and decoy strategies, measures of active defense, were introduced to secure both the proactive security and reactive adaptability of internet-of-things (IoT) networks that have been explosively applied to various industries without any strong security measures and to mitigate the side effects of threats. However, the existing MTD and decoy strategies are limited to avoiding the attacker's reconnaissance and initial intrusion attempts through simple structural mutations or inducing the attackers to a static trap based on the deceptive path and lack approaches to adaptively optimize IoT in consideration of the unique characteristic information by the domain of IoT. Game theory-based and decoy strategies are other options; however, they do not consider the dynamicity and uncertainty of the decision-making stages by the organizational agent related to the IoT domains. Therefore, in this paper, we present a type of organizational deception modeling, namely IoT-based organizational deception modeling (IoDM), which considers both the dynamic topologies and organizational business fingerprints customized in the IoT domain and operational purpose. For this model, we considered the practical scalability of the existing IoT-enabled MTD and decoy concepts and formulated the partially incomplete deceptive decision-making modeling for the cyber-attack and defense competition for IoT in real-time based on the general-sum game. According to our experimental results, the efficiency of the deceptive defense of the IoT defender could be improved by 70% on average while deriving the optimal defense cost compared to the increased defense performance. The findings of this study will improve the deception performances of MTD and decoy strategies by IoT scenarios related to various operational domains such as smart home networks, industrial networks, and medical networks. To the best of our knowledge, this study has employed social-engineering IoT knowledge and general-sum game theory for the first time.

**Keywords:** defensive deception; internet-of-things; moving target defense; decoy; game theory

## 1. Introduction

As the utilization of generalized heterogeneous IoT systems increases throughout various industries, sensitive sensor data and high authorities are frequently abused in static communication environments without applying any separate security measures, and the vulnerable attack surfaces of interconnected passive IoT systems and attacker dominant asymmetries continue to increase [1,2]. Most cybersecurity problems that occur within the IoT systems and networks are due to unique characteristics such as the passivity and heterogeneity of IoT devices, dependence on decision processing, low computational processing and resource allocation functions, and environmental hostility related to the application of existing security solutions [3–5].

As a lightweight counter-measure to mitigate the spatiotemporal defender inferiority issue of IoT systems, and to protect and defend IoT systems against potential advanced persistent threat (APT) attacks, defensive cyber deception techniques [6–8] were introduced. Defensive cyber deception techniques are non-cooperative decision-making pollution techniques that mislead potential attackers' cognitive perspectives, deceiving the attackers

into continuously composing and maintaining erroneous ex-post-action strategies through defender-dominant information asymmetry. These techniques have dedicated kill chain processes according to the operating environment and scenario along with unique characteristics different from other security techniques such as inducing, isolation, back-tracking, and mutation [9,10]. By applying these techniques to IoT systems and related networks, the spatiotemporal IoT attacker dominant asymmetry remaining in conventional security systems such as access control and intrusion prevention can be greatly alleviated [11], and flexible application and distribution of the techniques within multiple operating layers are facilitated by a small number of resources without large-scale changes to the system architecture and monitoring solution that have already been designed.

### 1.1. Background of Cyber Deception

Defensive cyber deception techniques are classified according to operational goals and their purpose of defense. They include MTD [12], Honey-X [13], and decoy [14,15]. In particular, MTD extremely restricts the validity of the defender surface information collected in advance by attackers based on cyber mobility properties such as shuffling, shifting, diversity, and redundancy. It is carried out toward periodic or non-periodic mutations of observable attack and exploration surfaces [16] according to the intention of the defender. In this way, MTD prevents the chain configurations by stage of attack chains toward an increase in the attacker's uncertainty and cognitive disturbance while resolving the defender's inferior information asymmetry issue, thereby realizing a proactive defense. Decoys and Honey-X are passive trap or active sandboxing entities that perform inducing and isolation to disturb attackers' cognition so that they attack false targets rather than the targets protected by the defender. Alternatively, they interact with the attackers so that the attackers are induced and isolated according to the intention of the defender.

Previous studies macroscopically combined MTD with decision strategies and learning theories, such as game theories [17–20], Markov decision process (MDP) [11,21–24], reinforcement learning, and adversarial attack-based machine learning schemes [25–34], intend to optimize benefits between attacks and defenses, thereby being strategized in order to always use optimized mutation strategies and diversify deception thresholds while attenuating the intrusion influences by the stage of the cyber kill chain (CKC) [35] and vulnerable surfaces. On a micro level, MTD was combined with other elements such as Honey-X and decoys, thereby being conceptualized considering diversified mutation items. In addition, MTD is a potential "game-changing" security solution in special domains such as cyber–physical systems, autonomous vehicles, smart factories, and smart grids that have unprotected communication characteristics [3,36–40].

### 1.2. Problem Statement and Related Limitation

However, the existing MTD and decoy concepts proposed to secure lightweight security in the IoT system have the following limitations.

- Unconceptualized deception strategies by various IoT domains: Existing defensive deception studies have conceptually formalized some of the operational characteristics and domain scopes of the target topology according to common vul-nerabilities and exploits (CVE) labeling in the national vulnerability database (NVD) for each IoT or non-IoT-based host. Accordingly, the related defense efficiency for each IoT-enabled deception strategy was derived. However, these studies were conducted only within a limited range, focusing on general-purpose fields such as WiFi-based smart home IoT and wireless sensor networks. In other words, it is reported that no systematic defensive deception strategy has been established for special domains such as medical IoT or industrial control systems.
- Unquantified organizational unique characteristics and IoT operation strategies: Previous studies have statically or less dynamically defined attack and defense goals, equilibrium states, decision weights, and state-transition probabilities based on common vulnerability scoring systems (CVSS) for IoT or non-IoT systems when construct-

ing scenarios and deception sequences. Accordingly, the related effectiveness and utility of the defender's deception based on the vulnerable contact points and attack vectors were calculated. However, the domain definition for the target IoT was limited only based on the related CVE vulnerability of the IoT device belonging to the topology. In addition, interactions between IoT devices, feedback, sequences, transmission/reception processes, and the organization's unique IoT operation strategy concept were not taken into account.

- Low flexibility and scalability issues of existing deception techniques: The naive MTD of previous studies was constructed to secure proactive defense by concentrating only on avoidance through mobility properties. In addition, it did not conceptually consider the reactive response for each threat that evaded the naive MTD scheme with sophisticated attackers. Other related studies with decoys have also not improved the scalability of the concept of deceptive perturbation by combining decoys with organization-specific characteristics.

- Unconsidered standardization of general sum game-based IoT deception strategies: Most of the defensive deception techniques applied in IoT, especially the strategic studies conducted to improve the defense efficiency by optimizing the MTD, have reached equilibrium states based on a zero-sum game. However, all cyber-attack defenses in the real world cannot be simulated solely on the basis of a zero-sum game. Accordingly, by introducing the concept of general sum game competition, preemptive normalization and optimization of non-zero-sum-based IoT-enabled MTD are also required.

- High dynamicity and real-time concepts that have not been applied: All deceptive disturbances and contingencies within an IoT network with defensive deception affect the organization's security posture and radically change the attack–defense environment. However, previous studies did not consider these factors and modeled the relationship between defender and attacker sequentially, mainly based on a simple leader and follower. For this reason, the proposed game models do not accurately characterize the real-time performance change of the actual IoT network attack–defense process and do not take into account the noise that can potentially affect the reward in each episode. In addition, even when performing steps to achieve goals for each actor within the game model, it is also impossible to dynamically calculate suboptimal deception strategies against incompletely perceived opponents according to sequential consumption of episodes for equilibrium [41,42].

*1.3. Research Goal and Key Contributions*

Therefore, in this paper, to supplement the existing MTD and decoys related to IoT security, we propose IoDM, an IoT-based organizational deception game model based on a perfect Bayesian Nash equilibrium (PBNE) and Bayesian stochastic Stackelberg game (BSSG)-based general sum game foreground and a partially observable Markov decision process (POMDP) state-transition background. Furthermore, to induce the attacker's initial cognitive bias according to the IoT defender's intentions and goals and secure the inferiority of non-deterministic reasoning, disinformation-based partial signaling and push behaviors are simulated, thereby making adaptive Bayesian decision-making for deceptive perturbation formulation undergo many steps. Thereafter, the IoT organization scenario based on the unique architecture and domain is formulated and simulations are compared based on the cyber kill chain (CKC) sequence, the defensive deception metric, and the decision parameters starting from the vulnerable point of contact to finally derive optimized results.

In this case, the following major contributions can be derived through this study.

- General sum-based IoT deception strategies can be additionally formulated: From the perspective of macroscopic strategization of defensive deception for IoT security, a general sum game-based competition concepts endowed with PBNE, BSSG, and partial signaling can be formed. That is, to expand the security of real IoT devices,

systems, and networks, the concepts of IoT-enabled MTD and decoys optimized for non-zero-sum games that were not reflected in existing zero-sum games can be compared and verified.

- Deception efficiency can be calculated by IoT domain and scenario: Through the topology templates configured for general purposes based on smart home IoT, cyber–physical system IoT, and medical IoT and related general sum game foreground components, real-time attack–defense competitive acts within an IoT organization with limited resources can be simulated and a multi-step spatiotemporal deceptive decision-making process configured. Furthermore, through the POMDP background components, the views not agreed upon between the attackers and the defenders by episode, the attack-exploration surfaces, and the concept of initiative can be also simulated. Moreover, IoT defenders can push disinformation behaviors for certain IoT devices and systems with the possibility of initial intrusion and final occupation according to a pre-calculated indicator of vulnerability (IoV) based on signaling. Furthermore, IoT defenders can dominantly conceptualize the cognition disturbance and social-engineering additionally defined in IoT-enabled MTD and decoys.
- A lightweight independent IoT deception defense concept can be established: When responding to specialized attackers, a concept of defensive deception can operate independently of the interconnected IoT systems without separately considering the deployment and management of dedicated protocols or additional solutions while minimizing resource usage and performance degradation and improving security can be defined based on IoT domains, scenarios, sequences, metrics, and major simulation parameters.
- Following the calculation of deceptive intensity, optimized deception strategies in the IoT domain can be produced: By subdividing functional parameters related to MTD and decoy performance (e.g., mutation target and period, shuffling, sampling scheme, decoy element, decoy properties, decoy path, maximum CKC permissible stage, target points by actor, and quantified main purpose), the most appropriate deception processes can be configured. In addition, an optimization strategy that can respond to the overhead problem and resource allocation issue of the existing IoT deception study cases where the concept of defensive deception was naively applied only for the purpose of simply improving security can be presented.

### 1.4. Structure of Paper

The rest of this paper is structured as follows. Section 2 compares and analyzes previous studies of MTD and decoys, which are the basis of this study, conducted based on game theory. Section 3 describes the IoDM based on PBNE, BSSG, and partial signaling-based general sum game foreground and POMDP state-transition background to derive MTD and decoy deception strategies optimized for IoT, and expatiates related games and MTD, decoy-related metrics and formulas, parameters, and IoT domain-based social-engineering knowledge. Section 4 provides formulated scenarios based on all of the IoT domains and topologies specified in IoDM, vulnerable points of contact, and related deceptive sequences. Furthermore, related attack–defense competition simulations are carried out to compare and analyze the results by metric. In Section 5, the threat-to-validity of this study and related improvement measures are discussed, and finally, in Section 6, conclusions are drawn.

## 2. Related Work

According to the recent trend of defensive deception studies, MTD partially falls within the scope of defensive cyber deception. However, unlike Honey-X, decoys, and fake objects, MTD neither disinforms nor artificially projects false information to actively mislead attackers. Instead, MTD conceptually involves inducing attackers' cognitive biases as intended by the defender. In addition, MTD does not neutralize attackers' observation and identification behaviors per se, such as obfuscation. Instead, MTD is an active-defense

paradigm that efficiently diversifies or proactively avoids the configuration of the variables of internal networks and heterogeneous system hosts by the domain to be protected, while maintaining the availability of major service functions provided to legitimate users and increasing the confusion or uncertainty of attackers, thereby preventing the formation of a successive attack chain. On the other hand, Honey-X and decoys perform reactive induction and isolation by manipulating, abusing, or misleading the attacker's cognitive aspects. Obfuscation is another separate study area, and its crucial difference is that it is closely related to the achievement of security goals at the data level [6].

Based on the room for controversies and study goals, the scopes of previous studies included 'Game-Enabled Defensive Deception Techniques with MTD' and 'MTD-based Defensive Deception Techniques for IoT', and the relevant techniques will be referred to in order to upgrade them into applied concepts for the improvement of deceptive performance and optimization in the IoT domain through the proposed IoT deception model and the general sum game strategy.

### 2.1. Game-Enabled Defensive Deception Techniques with MTD

The core of game theory studies using MTD is to model attackers' CKC tactics and the defenders' proactive evasion tactics based on shuffling and shifting in order to independently achieve the various goals possessed by individual actors, as well as to microscopically optimize the MTD variables such as mutation periods, mutation targets, and sampling functions by time point. In addition, macroscopically upgrading MTD strategies to maximize defenders' gains by minimizing performance degradation and maximizing security, and to minimize attackers' gains such as lateral movement and successful capture of the target host is also a main goal. Characteristically, the previous studies can be divided into 'general game-theoretic studies', 'Bayesian Stackelberg game-theoretic studies', and 'stochastic game-theoretic studies' [6,12,13].

Among the general game-theoretic studies, Zhu et al. [43] applied the sequential attack–defense competition formula based on two-person games and related metrics to the concept of MTD mutation, thereby quantifying the trade-off relationship based on the defender's security enhanced by MTD, the degraded operational performance, and the resource allocation function. Ge et al. [44] proposed an incentive-compatible MTD game based on communication mapping between normal users during server migration in order to guarantee high service visibility and throughput for legitimate users, and also characterize cyber agility [45] to secure additional availability. Neti et al. [46] configured an anti-coordination game as a guide framework for quantifying diversity-based deceptive measures in MTD and observing the interactions between scalabilities. Wright et al. [47] performed a two-player empirical game theory analysis to optimize the pre-conditions, required parameters, and target stability criteria for the formation of active MTD strategies against adaptive DDoS attackers. Carter et al. [48] proposed a dedicated MTD game architecture to secure migration optimization strategies in order to maximize seamless connections to legitimate users' services while minimizing the suspicion of induced and isolated attackers in the sandbox. On the contrary, Colbaugh and Glass et al. [49] argued that uniform randomization is an optimal strategy for diversity-based MTD.

Among the Bayesian Stackelberg game-theoretic studies aimed at optimizing the results of followers' behaviors according to the leader's behavior, Hasan et al. [50] proposed the co-resident attack mitigation and prevention (CAMP), a Nash balance game model that minimizes the ripple effect of internal and external threats in the intruded joint virtual environment while detecting the attacks of co-residents in the virtual environment sharing the same spatiotemporal resources. Feng et al. [51] proposed an artificial information disclosure model based on the Bayesian Stackelberg model that improves the agility of the defender by disturbing and partializing attackers' initial decision-making with the defender's intentional disclosure of false information. As a follow-up study, Zhu et al. [52] proposed a Stackelberg game framework that improves the efficiency of the attacker induction and isolation mechanism based on routing. The framework also created and disclosed

false packets specialized in the reconnaissance stage of internal and external attackers so that the directivity of application of the concept of deception according to the composition of a scenario model close to the practical environment can be secured. Sengupta et al. [53] proposed a Bayesian Stackelberg game-related model for the placement of IDS solutions on the Web and cloud to organize an MTD strategy that maximizes the proactive security using the system configuration set candidate group while minimizing the mutation cost and performance degradation rate of the defender with limited resources. Another study conducted an MTD strategy study for zero-sum game competition based on general-sum games to secure defensiveness against APT attacks in cloud networks [54]. As a follow-up study, Li et al. [55] proposed a Markov Stackelberg game together with optimization formulas based on the average-cost semi-Markov decision process (SMDP) and the discrete time Markov decision process (DTMDP) to produce the defender's spatiotemporal MTD mutation decision-making process against advanced attackers. Seo et al. [56,57] proposed an active cognitive disturbance function not influenced by the existing MTD concept and combined it with a social engineering decoy sandbox layered in the form of organizational open-source intelligence (OSINT) to form defensive deception concepts optimized for actual organizational operational goals. Based on these previous studies, a real-time attack–defense competition in an organizational environment with limited resources could be simulated and multi-staged deceptive decision-making processes could also be modeled depending on the scenario. In addition, the limited fields of view by the actor and the manipulated present situation of host occupancy based on the reward values by episode could also be simulated with partial signaling based on disinformation.

Among stochastic game theory studies that can reflect the relationships between multiple players through stochastic transfer, Manadhata et al. [58] proposed a game model that diversifies the dynamics between attacks and defense based on the concept of stochastic transfer according to the flow of decision-making and reflects the diversified dynamics on strategies by MTD mutation state. Based on the previous studies, the MTD balance concept and related trade-offs could be formulated to model the optimal proactive defense strategies based on the dynamic surface compositions by scenario environment. To deal with the incomplete information by an actor in the MTD game model, Zhang et al. [59] proposed a Nash-Q learning algorithm based on a reward matrix constructed by capturing the frequency and distribution of an attacker's strategy selection. The fact that the Nash theory can better reflect actual organizational operation scenarios compared to other game theories could be derived and the trade-off quantified based on the availability of legitimate users according to the calculation of MTD-based deceptive items could be calculated.

### 2.2. MTD-Based Defensive Deception Techniques for IoT

IoT-enabled MTD has not been performed universally due to the inherent static characteristics and performance limitations of the existing IoT systems. However, recently it has been performed with an aim to characterize shuffling tactics of IoT features, improve cyber resilience at the protocol level, and optimize the deceptive strength through zero-sum game-based strategies. These studies are typically divided into 'definition of MTD strategies for IoT' and 'construction of IoT framework with MTD'.

Among the studies of the definition of MTD strategies for IoT to optimize the MTD strategies according to the unique characteristics of the IoT system, Navas et al. [60] proposed a strategy to randomize the spreading sequences in the direct-sequence spread-spectrum (DSSS) system based on cryptographically secure pseudo-random (CSPR) in order to minimize the ripple effect of potential internal–external jamming attacks on DSSS communication in heterogenous IoT systems. This strategy could be evaluated and verified to be more energy-efficient compared to the existing anti-jamming techniques and to improve the defense of the communication service in ad hoc IoT networks. Ge et al. [61] proposed a concept of IoT decoy induction pathways, a network topology shuffling-based MTD (NTS-MTD) strategy that performs shuffling on real IoT networks to minimize the spatiotemporal dominance of asymmetric IoT attackers as well as concretizing the deception

strategies optimized for static IoT systems in organizations that have only limited available resources by the MTD mutation metric. Based on the previous strategy, the levels of overall deception efficiency by scenario could be compared and verified in relation to all of the MTD mutation periods, items to be mutated, mutation sampling techniques, and specified IoT single or multiple operation goals, and then analyzed based on IoT performance, and down-time, and network security could be derived. Nizzi et al. [62] proposed an address shuffling algorithm (AShA), an algorithm that utilizes hash-based message authentication code (HMAC)-based cryptographic hash to realize rapid MTD shuffling at the MAC-IPv4-IPv6 level in the IoT network while minimizing the overhead. Through the AShA, the functional stability related to MTD updates in a network composed of numerous IoT nodes could be derived based on collision, and the performance could be evaluated based on a Raspberry Pi and Carambola. As another study focused on mutating the IP address of IoT systems, Zeitz et al. [63] proposed a micro-moving target IPv6 defense (μMT6D) as an MTD mechanism based on a lightweight hashing algorithm dedicated to low-power IoT. According to the study, μMT6D could be proved to be robust against DDoS or eavesdropping passive attacks in IoT environments.

Among the studies on the construction of an IoT framework with MTD for the composition of guidelines defined for the design of IoT-based MTD strategies, Navas et al. [64] proposed IANVS as a universal framework that supports the intensive design, implementation, and evaluation of MTD strategies for IoT systems. This proposed framework is related to the interaction between the gateway and nodes and concretized MTD strategies such as UDP port and CoAP protocol-based shifting targeting real IoT hardware devices. According to the study, an architecture where the common components of the MTD strategy were abstracted, generalized, and interconnected could be designed, and the efficiency of proactive defense against DDoS attacks could be evaluated as probabilities by the CKC step. Kyi et al. [65] proposed the directivity of proactive framework designs related to IP address shuffling in the IoT communication layer and code diversification in the data layer. Mercado et al. [66] randomized the communication protocol between nodes and gateways in the IoT network to propose an MTD strategy and an architecture that minimizes the asymmetric spatiotemporal dominance of attackers and the defender's system performance overhead based on multiple-criteria decision analysis. Based on the previous studies, using the MTD strategy parameter, we could prove the levels of the efficiency of proactive defense against DDoS attacks targeting IoT networks and the conceptual approaches for future optimization based on game theories and genetic algorithms.

### 2.3. Taxonomy Analysis by Previous Studies for Proposed Model

Out of the above-mentioned related works, refs. [18,20,53,54,56,57] among previous game theory studies and [64,66] among previous IoT-enabled MTD studies provided the main inspirations for this study. Accordingly, to alleviate the above-mentioned limitations, we proposed IoDM, a systematic IoT deception model based on PBNE and BSSG and a partial signaling-based general sum game foreground and a POMDP state-transition background. Its related internal components, processes and schemes, and taxonomic analysis are presented in Table 1.

**Table 1.** Taxonomy of existing defensive deception research studies and proposed deceptive model for IoT.

| Approach | Specific Technique | Advantages | Disadvantages |
|---|---|---|---|
| Game-Enabled Defensive Deception Techniques with MTD [43–59] | General game, Bayesian Stackelberg game, Stochastic game | Formalizing the interactions between attackers and defenders and providing methodologies for estimating deception effectiveness and security. Determining the optimal MTD strategy based on learning and realizing the decision model. | Not working the best strategy devised by defender for the irrational attacker. Requires very high number of deriving computational resources for the optimal solution space. Causing uncertainty because of interpreting the same game differently depending on their subjective perception of different players. Difficulty in modeling cybersecurity and defensive cyber deception issues in real-world and practical environments. |
| MTD-based Defensive Deception Techniques for IoT [60–66] | Definition of IoT-enabled MTD strategy, Construction of IoT framework with MTD scheme | Securing an avoidance-based, independent-adaptive mobility-type countermeasure for attenuation of the explosively increased attack-detection surfaces in heterogeneous IoT networks. Breaking down the spatiotemporal dominance of attackers in static and passive IoT systems. Reducing organizational operating costs for deploying security solutions dedicated to IoT. Adopted as the most core security game-changer for special IoT domains with insecure communication channels | MTD performance constraints occur due to limited and non-uniform resource allocation within IoT-based organizational environments. Due to the characteristics of interconnected heterogeneous IoT communications, the consistency of the efficiency of MTD-based proactive defense cannot be maintained. Not suitable for extremely lightweight IoT communication domains. Not mapped to free IoT node joining and leaving processes |
| **IoDM** | - Defensive deception technique for IoT<br>- IoT-enabled organizational MTD, IoT-enabled organizational decoy<br>- Game-theoretic technique for IoT<br>- Perfect Bayesian Nash equilibrium (PBNE) with general-sum, Bayesian stochastic Stackelberg game (BSSG), Partially signaling scheme for disinformation (PSG)<br>- Decision theoretic technique for IoT<br>- Partially observable Markov decision process (PDMDP) | **Description and Improvement**<br><br>The IoT-enabled organizational MTD is the concept of using an MTD strategy to select appropriate mutation periods and intensities, and targets by IoT domain. Furthermore, this MTD determines the sampling techniques and shuffling schemes with the advantage of the defender. Through the definition of these adaptive IoT-enabled MTD, optimized MTD strategies can be secured by IoT domain and related mutation metric, and reactive tactics within the MTD for the defender to carry out in-depth deception can be realized.<br>The IoT-enabled organizational decoy is a concept of induction of CKC attacker cognitive disturbance consisting of layered false organizational information based on certified IoT CVE and IoV information, which standardizes the decoy elements, properties, and targets based on unique characteristics by IoT domain, and optimizes the decoy induction pathways allowable values. Through the definition of active IoT-enabled decoys, the usability and scalability of the decoy sandbox in the IoT environment can also be improved.<br>The organizational IoDM model is a two IoT player competitive game mainly composed of PBNE, BSSG, partial signaling-based general sun game foreground components, and POMDP state-transition background components and a real-time attack–defense process within an IoT organization operating environment with limited resources was representatively configured as multi-staged spatiotemporal deception decision-making. | |

## 3. Proposed Organizational Deceptive Modeling for IoT and Related Strategies

In this section, IoT-enabled MTD and decoys considering organizational IoT knowledge are formulated. The major components and detailed modules in the proposed IoDM are configured, and all the overall deception-game metrics related to PBNE, BSSG, signaling, and POMDP and formulas are also defined.

### 3.1. Design Principle

Figure 1 represents the major architecture of the proposed IoDM. First, in a social engineering deception knowledge for IoT components (①), organized IoT elements and operations are applied as deceptive knowledge to be used for IoT-enabled MTD and the decoy. Thereafter, through the deceptive knowledge, the properties, functions, and

parameters in the MTD and decoy are defined, formulated as a deception process, and used for the dynamic game-based foreground components and the state-transition-based background components. In the dynamic game-based foreground components (②), PBNE, BSSG, and partial signaling decision techniques are all mixed in the game module as variables for generating, updating, and signaling the deception tactics of the IoT defense actors. They are also the major schemes for configuration of the counter-measure tactics of the IoT defense actors in the attack–defense module. The attack/defense sequences by actor are then established, and PBNE, BSSG, and partial signaling-based competitive games are played. Next, in the state-transition-based background component (③), to continuously maintain the asymmetric dominance of the IoT defender, all the deceptive variables for the deceptive signaling sequence based on disinformation, artificial disclosure, deceptive perturbation for cognitive disturbance, occupancy manipulation, etc. and CVE- and IoV-based vulnerability determinants within transition metrics are applied. Thereafter, based on the configured scenario template and matrix, the POMDP is formulated and the decision actions by the actor are schematized. Finally, IoT-enabled MTD and decoy-based deception strategies, competition between general sum-based attack–defense actors and related reward concepts, payoff tactics, and state entry conditions for game equilibrium are all considered to derive results.

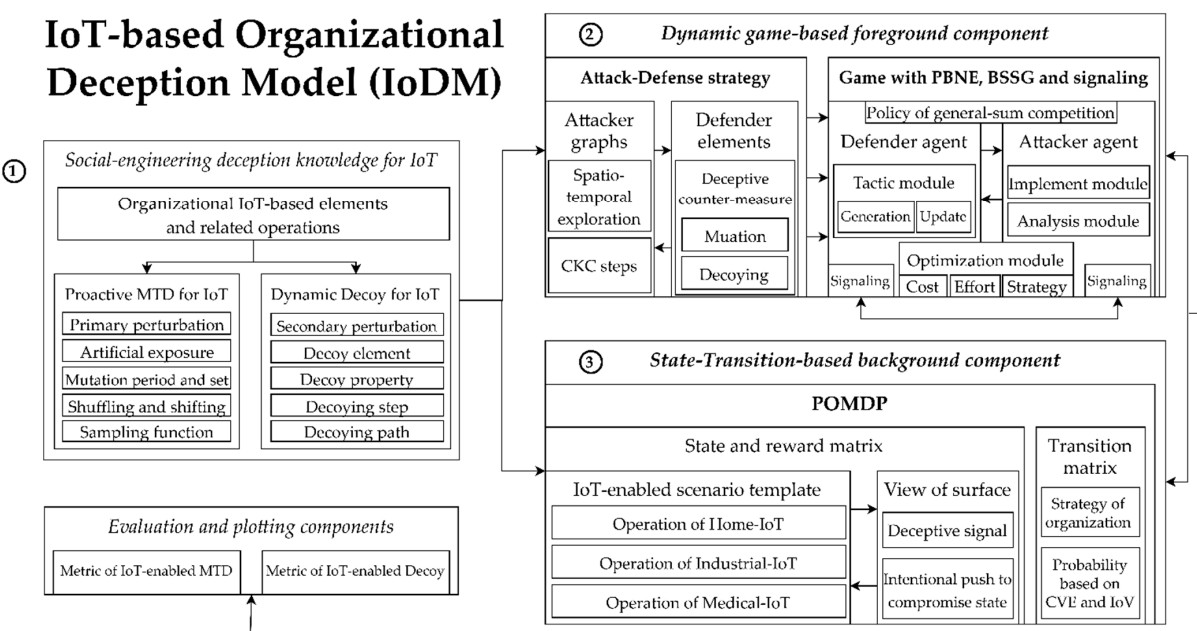

**Figure 1.** The main overview of the proposed deceptive approach for IoT-enabled organizational environment.

### 3.2. IoT-Enabled MTD and Decoy-Based Deception Process

The IoT-enabled organizational MTD applied within IoDM is a concept of using the MTD strategy to select appropriate mutation periods, intensities, and targets by IoT domain, and determine sampling techniques, shuffling schemes, etc. with advantages to the defender. That is, depending on the acts of the external IoT attacker who intends to invade and occupy internal IoT devices, systems, and subnetworks, mutation strategies are selected, and the intensity of evasion is adjusted under the intervention of the defender to disturb the cognitive judgment of the attacker so that wrong defender intelligence is constructed based on passive deception perturbation [13] or active disinformation. According to previous studies, reactive tactics can be additionally applied within the MTD for the defender to carry out critical system-based in-depth deception while proactively configuring mutation tactics optimized in the IoT domain and related mutation metric.

The IoT-enabled organizational MTD is therefore defined to be centered on the Bellman value iteration-based IoT-based mutation sampling (VS) for adaptive MTD configuration according to changes in attack actions together with the perturbation (P) to asymmetrically assign noises to the attacker's cognitive directivity and is configured as shown in Equations (1) and (2), respectively [56,57]:

$$P = Pr[L = l | OS = \mu(os)]. \tag{1}$$

In this case, $L$ is a set of virtual elements similar to the unique fingerprint of the actual IoT device and system to be protected in order to distort the bias composition gradient of the attack surface-based intelligence identified by the IoT attacker at the present time in favor of the IoT defender. However, it is hierarchically configured based on the IoT network and host layer so that only legitimate users can identify it, and it is dynamically signaled under the leadership of the defender based on the internal rule table. The $OS$ is a set of actual IoT specification information groups that are referred to in order to minimize attackers' suspicion about the defender's maneuver based on artificial disclosure and disinformation, and acts to configure deceptive perturbation through $L$ while improving the deception efficiency of the generated $L$ and $L`$. The $\mu(os)$ is a function to calculate the possibility for the attacker to predict the entire secrets being protected by the deceptive defender considering the cognitive ripple effect of $os$, the elements of the IoT specification information partially observable by the attacker, along with $L$ and $L`$.

$$VS^n(i) = \min_{\tau_i, m_i \in M} \left[ c_{i,j} + \sum \widetilde{m}_{i,j} VS^{n-1}(j) \right], \tag{2}$$

$c_{i,j}$ is the cost of mutation to shift the surface index $j$ of the defender in the next episode with incomplete private information through the adaptive deceptive signaling action of the IoT defender in the current episode in the IoT defender's surface index $i$ in order to optimize the deceptive trade-off and minimize the overhead in an IoT-based organizational network having limited operational resources. $\widetilde{m}_{i,j}$ represents the possibility for external IoT attackers following changes in $i$ and $j$, and is configured to minimize the suspicion of the attackers who continuously search for the target IoT device, system, and network status. $\tau_i$ is the spatiotemporal cost consumed by the defender to maintain dominance until the completion of mutation of the IoT surface elements for $i$ and the mutation time slot length and $m_i$ is the IoT surface element sampled and optimized based on $i$ within the deceptive surface set $M$ generated through mutation.

IoT-enabled organizational decoy is the concept of a multitenancy type dynamic sandbox that extremely limits the attacker's CKC-based ripple effect by actively inducing cognitive disturbance of the IoT attacker. Alternatively, it isolates the attacker in a hole from which escaping is difficult based on all of the CVE vulnerabilities, CVSS scores, IoV information, and actual IoT specification information disclosed by major IoT domains in order to further improve the low deceptive IoT defense efficiency of the existing decoy. That is, it protects the actual IoT device, which is the ultimate target of intrusion of the attacker, from exposure to the attacker, while standardizing all decoy elements and properties, and targets based on IoT as if they are valuable objects of protection and optimizing decoy induction pathways and allowable values in order to immediately seduce the attacker. Concretely, based on detailed decoy attributes such as the 'distinguishability' of the decoy that resembles the object of protection from the outside but is distinguished as an actual false dummy by internal legitimate users with separate prior knowledge, 'dazzling' to induce intrusion by an unidentified external attacker, 'enticing' to actively entice identified attackers, 'redundancy' and 'diversity' to minimize attackers' suspicion, 'detectability' to identify attackers deceived by the decoy based on the beacon, and 'controllability' to set multiple levels of the degree to which attackers' intrusion and escape are allowed according to weights, the dynamic actions to gradually induce the attacker's inferiority and force the attacker to follow defender's intentions are continued. Active decoy tactics optimized in

the IoT domain and related decoy metrics can be reactively constructed while expanding the usability and scalability of the decoy sandbox in the IoT environment. In addition, when the IoT-enabled organizational decoy is combined with IoT-enabled proactive MTD performed primarily at the IoT network and host layer, asymmetric inferiority can be secondarily forced on external IoT attackers who bypassed MTD-based evasion. That is, cyber agility within major IoT networks is ensured by the IoT-enabled proactive MTD, and cyber resilience is ensured by the IoT-enabled reactive decoy.

The attributes in the IoT-enabled decoy are centered on the decoy principle [15], and are configured as shown in Equations (3)–(8), respectively [56,57]:

$$Pr\left[Exp_{A,\,H,\,O}^{believe} = 1\right] \leq \frac{1}{2}. \tag{3}$$

Equation (3) is believability (B), which calculates the attack probability of attacker $A$, who has no prior knowledge to distinguish decoy set $H$ from the actual object of protection, using $O$, which is a set of organizational IoT specification information element groups reconstructed for IoT-enabled decoy operation. That is, when $A$ constitutes the attack surface at the present time, B judges $H$ as a decoy of the defender and ensures that the probability to exclude $H$ as noises is not higher than 0.5.

$$Pr[o \rightarrow O | o \in PF] = Pr[h \rightarrow O | h \in H]. \tag{4}$$

Equation (4) is enticing (E), which utilizes $o$, IoT specification information elements for decoy operation in $O$, and $PF$, an indicator of the degree to which an IoT attacker's hasty judgment is quickly enforced in favor of the defender, to calculate the possibility of dynamic signaling that will actively induce the act of intrusion of $o$-based decoy element $h$. Thereafter, E also verifies whether the deceptive attractiveness of the $PF$ in $h$ managed by the IoT defender is similar to the uniqueness of $o$ introduced to minimize the attacker's suspicion.

$$\prod_{i=0}^{n} Pr[V_i] > \delta. \tag{5}$$

Equation (5) is conspicuousness (C), which utilizes both the $i$-based intelligence constructed at the present time according to the signaling between actors and the view $V_i$ that can be observed by the other party as surface information to calculate the possibility of static signaling related to $\delta$, a dazzling indicator of elements that induce extreme bias in the initial cognition of the attacker such as CVE vulnerability in $o$. In addition, while advertising CVE vulnerability in $o$ to the attacker, C also ensures $V_i$-based artificial information disclosure to prove false B about the vulnerability to the attacker.

$$Pr[h \rightarrow O : CD_{A,h} = 1] \geq \epsilon. \tag{6}$$

Equation (6) is detectability (DE), which calculates the detectability related $\epsilon$, the detection threshold in $h$ using $CD_{A,h}$. $CD_{A,h}$ is an indicator of the degree of detection of intrusion into $h$ by attacker $A$ for adaptive management of $h$ according to changes in IoT-based organizational networks. In addition, DE ensures that the possibility of false detection and non-detection based on decoy beacons is minimized.

$$Pr[CT_{D,\,o,\,h} = 1] = Pr[CT_{D,o,h} = 1 | H]. \tag{7}$$

Equation (7) is non-interference (NI), which calculates the possibility of adaptive non-interference for the situation where the contiguity of defender $D$ for $o$-based $h$ was clearly distinguished as a leader for deceptive signaling and was authorized not to be decoyed.

That is, NI ensures both role-based access control and controllability so that attacker *A* can access only *h* and defender *D* can access the actual object of protection.

$$Pr\left[Exp_{D,H,O}^{believe} = 1\right] = 1. \tag{8}$$

Equation (8) is differentiability (DI) based on Equations (3), (6) and (7), which calculates the possibility for defender *D* to distinguish between *O* based *H* and the actual object of protection and the impossibility for attacker *A* to distinguish the same. That is, DI is a reactive secondary deception policy for attackers who successfully bypassed IoT-enabled proactive MTD, which ensures that the attackers cannot distinguish between *H* and the actual object of protection.

Finally, all of the IoT-enabled MTD and decoys are upgraded into deceptive processes in the IoT-based organizational networks by the major IoT domain and utilized as atomic variables that are required without fail for all competitive actions using all these IoT-enabled MTD and decoys as deceptive processes in PBNE, BSSG, and partial signal-based dynamic game foreground components and POMDP state-transition-based background components.

### 3.3. Construction of Deceptive Game Architecture with IoT-Based Organizational Network

In this section, we define the main tuples, equations, and related metrics for the dynamic game-based foreground components and state-transition-based background components configured in IoDM.

#### 3.3.1. Regularization of General-Sum Game Mechanisms

As shown in Figure 1, the dynamic game-based foreground components typically consist of a PBNE, BSSG, partial signaling-based general sum game competition module and an attack–defense strategy module to define the state-transition probabilities by the actor. As shown in Figure 2, the general sum game competition module adopts all of the PBNE decision strategies to maximize the payoffs by an episode of private asymmetry based on incomplete information by attacker and defender, the BSSG decision strategy to optimize the quantitative sequential relationships of micro or macroscopically calculated reward values as an active leader and reactive follower-based causality, and the partial signal game decision strategy to actively force attackers' prior beliefs and confusion and maintain and sustain defender-dominant leadership. The attack–defense strategy module refers to all of the IoT devices formulated by smart home IoT, industrial IoT, and medical IoT-based organizational scenarios defined in advance in the IoT-enabled scenario template, systems, sub-farm networks, intercommunication channels, operational strategies, CVE vulnerabilities, and CVSS scores to formulate the target, tactic, and sequence-based attack–defense flows produced by the actor as threat modeling.

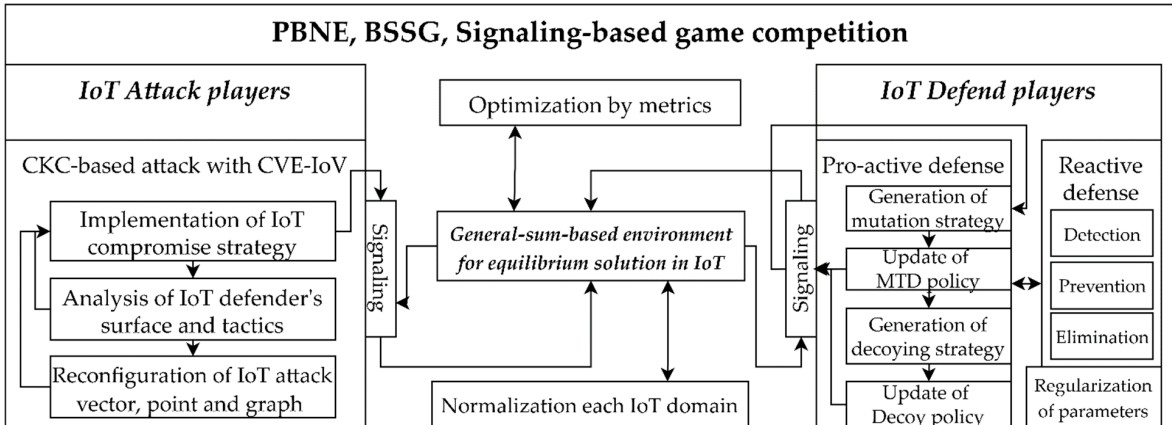

**Figure 2.** Detailed sub-overview of the general-sum-based attack–defense competition with PBNE, SSG, and signaling.

In this case, the dynamic game-based foreground components, as shown in Figures 1 and 2, are composed centered on the following 12-tuples.

- $N = (N_A, N_D)$ is a set of actors, where $N_A$ is an IoT attacker, and $N_D$ is an IoT defender. In this case, payoffs and signaling by the actor and the leader–follower relationship are defined differently depending on each IoT scenario and the present situation of the development of detailed episodes.

- $TS = (TS_{N_A}, TS_{N_D})$, $TS_{N_D} = (ts_i | i = 1, 2, \ldots, n)$, and $TS_{N_A} = (\rho)$ are the sets of elements uniquely possessed by actors. $TS_{N_A}$ is defined as the deceptive private information element of IoT defender $N_D$. $TS_{N_A}$ is defined as the attack graph-tree-based private information element of IoT attacker $N_A$. The elements are combined or divided according to the changed payoffs by the actor, and unlike the defender, the attacker dynamically composes the element with $\rho$, which is the effectiveness of the IoT attack surface at the present time point.

- $GS = (GS_{N_A}, GS_{N_D})$, $GS_{N_D} = (gs_{N_{di}} | i = 1, 2, \ldots)$, and $GS_{N_A} = \left( gs_{N_{aj}} \middle| j = 1, 2, \ldots \right)$ are the sets of decision strategies for general sum competition between IoT attacker $N_A$ and IoT defender $N_D$, which are also configured according to the payoffs and signaling by the actor and the leader–follower relationship. $GS_D$ is the deceptive strategy possessed by the defender. $GS_A$ is the defender surface information possessed by the attacker, and is defined as an attack graph-tree and intelligence-based strategy.

- $SS = (SS_{N_A}, SS_{N_D})$, $SS_{N_D} = (ss_{N_{di}} | i = 1, 2, \ldots)$, and $SS_{N_A} = \left( ss_{N_{aj}} \middle| j = 1, 2, \ldots \right)$ are the sets of the signals of IoT attacker $N_A$ and IoT defender $N_D$, respectively, and are selected according to the signaling initiatives given by the actor. $N_A$ has $SS_{N_A}$, which is a set of attack signals to achieve the goal of intrusion, and $N_D$ has $SS_{N_D}$, which is a set of proactive–reactive defense signals for achieving the protection of all IoT devices, systems, and networks.

- $\omega$ is a signal attenuation factor that determines the degree of attenuation of the $SS_{N_D}$) of IoT defender $N_D$ according to the progress of the episode.

- $GB = \left( GB_A, \widetilde{GB_A} \right)$, $GB_A = (GB_A(gs_{N_{di}}) | i = 1, 2, \ldots)$, and $\widetilde{GB_A} = GB_A(gs_{N_{di}} \cdot \omega)$ are the sets of general sum-based game beliefs of IoT attacker $N_A$. $GB_A$ is the set of prior beliefs of $N_A$ and $\widetilde{GB_A}$ is the set of posterior beliefs of $N_A$ produced through Bayes' rule after $N_A$ received normal signals or deceptive signals spoofed based on the $SS_{N_D}$ of IoT defender $N_D$.

- $S = (s_i | i = 0, 1, \ldots k)$ is a set of $GS$ and $SS$-based finite states in a general sum-based dynamic game component, which defines the multi-level nature and transferability of attack–defense competition along with actions.

- $A = (A_{N_A}, A_{N_D})$, $A_{N_D} = \left( a^i_{N_{d_i}} \middle| i = 1, 2, \ldots x \right)$, and $A_{N_A} = \left( a^j_{N_{a_i}} \middle| j = 1, 2, \ldots y \right)$ are the sets of finite actions of IoT attacker $N_A$ and IoT defender $N_D$ for $S$. $A_{N_D}$ defines the defender's proactive–reactive deceptive and defensive actions to $s_i$ as transition relations. $A_{N_A}$ defines the attacker's CKC actions to $s_i$ such as reconnaissance and search, vulnerability and fingerprint-based exploits, initial occupation and lateral movement, final invasion through privilege escalation and takeover and occupation.

- $\theta \left( S_k, a_x, a_y, S_{k'} \right)$ is a probability distribution function used to calculate the possibility for the IoT attacker $N_A$ and IoT defender $N_D$ to arrive at $S_{k'}$ when IoT attacker $N_A$ carries out the action termed $a_x$ and IoT defender $N_D$ carries out the action termed $a_y$ at $S_k$ in the current episode.

- $R \left( S_k, a_x, a_y \right)$ is a function used to calculate the reward obtainable within the current episode in when IoT attacker $N_A$ and IoT defender $N_D$ carry out the actions termed $a_x$ and $a_y$ at $S_k$, respectively. IoT actors compete toward the maximization of reward $R$ until entering general sum-based game equilibrium.

- $U = (U_A, U_D)$ is a discount factor function that cuts off the judgment ranges by the actor within [0,1] to force a quick judgment while also reducing the solution space required for optimization. In addition, it defines leader–follower-based ex-ante/ex-

post competitive strategy judgments such as the limitation of surface information by an actor that can be observed at the present time, intelligence distortion, disinformation, and artificial disclosure so that they can be indirectly simulated.

- $CU = (CU_A, CU_D)$ is a utility function for resources and costs incurred when performing competitive actions. $CU_A$ is the utility function of IoT attacker $N_A$, and $CU_D$ is the utility function of IoT defender $N_D$.

As specified in the 12-tuples as such, the action processes of the attacker and defender in each IoT scenario are defined based on the multi-layered information transmission-based partial signal game and the leader–follower causal relationship-based BSSG. The payoff optimization scheme is dynamically constructed with respect to game equilibrium in the PBNE.

### 3.3.2. Optimization of Attack-Defense Competition with Game Equilibrium

Through PBNE-based game equilibrium, the decision sequences for optimizing the reward values by an actor in the general sum competition relationship are divided according to the initiative in BSSG and partial signaling-based signal spoofing.

If an IoT attacker is selected as an active compromise leader from the viewpoint of attack actions by episode, and signals for CKC actions such as reconnaissance, search, exploitation, privilege escalation, and IoT system occupation are also transmitted as intended by the attacker, both the efficiency of proactive deception and the efficiency of reactive defense of the IoT defender, who is a passive follower and attack signal receiver, will decrease. Additionally, there will be the formation of a spatiotemporal asymmetric attacker-dominant relationship regardless of the defender's defense goal. Conversely, if an IoT defender is defined as an active deceptive leader from the viewpoint of defense actions by episode, and deceptive signals are transmitted to force defensive deceptive actions such as disinformation, artificial disclosure, deceptive perturbation, and cognitive disturbance, the possibility of success of the attacks by the IoT attacker will decrease. On the other hand, the attack cost and the rate of consumption of utility and resources will increase conversely to both the maintenance and sustenance of the dominance of the defender regardless of the attacker's attack goal. In such a competitive process, as the range of judgment of each actor is temporally and spatially cut off according to a predefined discount factor, the reduction of the solution space for calculation of the optimal values when the goals of the actor are achieved and the process of normalization to approximate values are both carried out a priori.

Within the leader–follower relationship based on signaling by episode, the general sum reward optimization scheme related to the dependent reasoning action according to the leading actor is organized as a $Q$-value as shown in Equation (9). In this case, $U$ and $TS$ are defined as actors with signaling initiatives in the current episode:

$$Q(S_k, a_x, a_y) = R(S_k, a_x, a_y) + U \sum\nolimits_{S_k} \theta(S_k, a_x, a_y, S_\kappa) \cdot TS \cdot OPT(S_\kappa) + CU, \qquad (9)$$

where $OPT(S_\kappa)$ from the viewpoint of the IoT attacker who actively carries out signaling is configured into Equation (10) through $SS$. $SS$ are signaling acts that can be performed in $S_k$, and GB. GB is a related belief and produces the optimized reward value with private information-based incomplete judgment:

$$OPT(S_\kappa) = \max_{SS} \min_{a_x} \sum\nolimits_{ay} Q(S_k, a_x, a_y) \cdot (ss_{N_{di}} | i = 1, 2, \ldots) \cdot GB. \qquad (10)$$

PBNE-based game equilibrium is determined based on $OD$ and $OA$ as shown in Equations (11)–(14). In this case, $P_D$ in Equation (11) similarly configured based on $OPT$ and $GB$ is the probability of prior probability-based judgment of the IoT defender. $P'_D$ *in* Equation (12) is the probability of posterior probability-based reasoning by the IoT defender related to $SS_{N_D}$ reconfigured based on the updated internal deception-defense strategy after the IoT defender's feedback-based signaling to $SS_{N_A}$. In addition, when

calculating the PBNE-based game equilibrium through $P_D$ and $P'_D$, the result is affected by $U$, *a* discount factor, and the time to reach the equilibrium in the game is controlled by the configuration of $U_A$, or $U_D$ related to the signaling initiative:

$$P_D = \left( p_D \cdot (TS_{N_{D_i}}) \middle| i = 1, 2, \ldots n \right), \tag{11}$$

$$P'_D = P'_D \left( \left( TS_{N_{D_i}} \middle| i = 1, 2, \ldots n \right) \middle| SS_{N_A} \right), \tag{12}$$

$$OD(SS_{N_{A_j}}) = arg \max_{SS_{N_{Dk}} \in SS_{N_D}} \sum_{TS_{N_{D_i}} \in TS_{N_D}} P'_D \cdot F(TS_{N_{D_i}}, SS_{N_{Aj}}, SS_{N_{Dk}}), \tag{13}$$

$$OA(TS_{N_{A_i}}) = arg \max_{SS_{N_{A_j}} \in SS_{N_A}} F(TS_{N_{A_i}}, SS_{N_{Aj}}, OD(SS_{N_{A_j}})). \tag{14}$$

Finally, to reflect the general sum-based attack–defense competitive game in IoDM based on state-transition, the POMDP-based state-transition background component specified in Figure 1 is structured into the form of an MDP-based interface as shown in Figure 3. The transition probability and compensation value in the POMDP shown in Figure 3 are defined as shown in Table 2 but they are calculated differently as shown in Tables A1 and A2 of Appendix A according to the major IoT scenarios defined in advance.

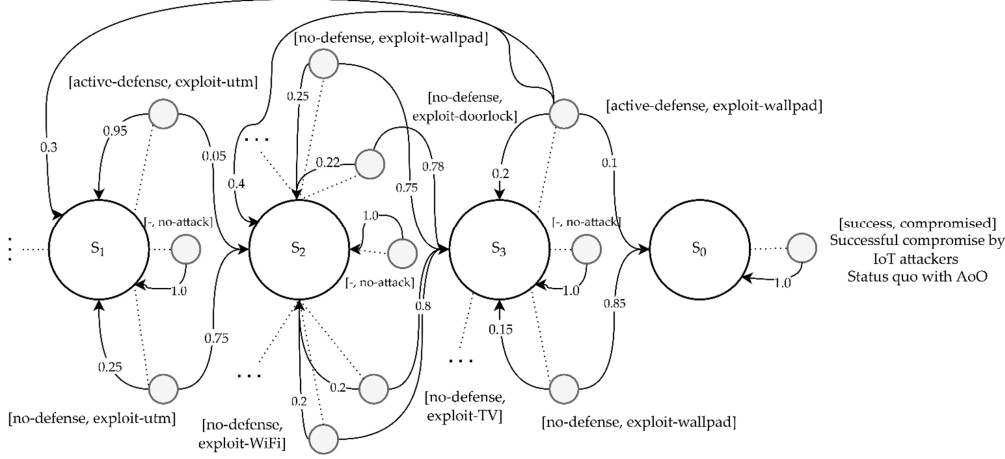

**Figure 3.** Detailed sub-overview of the POMDP with defender side in IoT-enabled Scenario 1.

**Table 2.** Probability matrix of transition and semi-constant reward value with payoff strategy in IoT-enabled Scenario 1.

| State | Probability of Transition [1] in Scenario 1 | Reward Value for Defender |
|---|---|---|
| $S_0$ | $[(1, 0, 0, 0)]$ | $[\ -50\ ]$ |
| $S_1$ | $\begin{bmatrix} (0, 1, 0, 0) & (0, 1, 0, 0) \\ (0, 0.25, 0.75, 0) & (0, 0.95, 0.05, 0) \end{bmatrix}$ | $\begin{bmatrix} 0 & -2 \\ -10 & 10 \end{bmatrix}$ |
| $S_2$ | $\begin{bmatrix} (0, 0, 1, 0) & (0, 0, 1, 0) & (0, 0, 1, 0) & (0, 0, 1, 0) & (0, 0, 1, 0) \\ (0, 0, 0.2, 0.8) & (0, 0.7, 0.3, 0) & (0, 0.2, 0.8, 0) & (0, 0.2, 0.8, 0) & (0, 0.15, 0.85, 0) \\ (0, 0, 0.2, 0.8) & (0, 0.5, 0.4, 0.1) & (0, 0.7, 0.3, 0) & (0, 0.3, 0.7, 0) & (0, 0.3, 0.7, 0) \\ (0, 0, 0.25, 0.75) & (0, 0.5, 0.4, 0.1) & (0, 0.3, 0.7, 0) & (0, 0.7, 0.3, 0) & (0, 0.2, 0.8, 0) \\ (0, 0, 0.22, 0.78) & (0, 0.5, 0.4, 0.1) & (0, 0.3, 0.7, 0) & (0, 0.3, 0.7, 0) & (0, 0.7, 0.3, 0) \end{bmatrix}$ | $\begin{bmatrix} 0 & -3 & -2 & -2 & -3 \\ -10 & 9.5 & -2 & -2 & -3 \\ -10 & -3 & 9.5 & -2 & -3 \\ -6.5 & -3 & -2 & 5.5 & -3 \\ -8.6 & -3 & -2 & -2 & 7.2 \end{bmatrix}$ |
| $S_3$ | $\begin{bmatrix} (0, 0, 0, 1) & (0, 0, 0, 1) \\ (0.85, 0, 0, 0.15) & (0.1, 0.3, 0.4, 0.2) \end{bmatrix}$ | $\begin{bmatrix} 0 & -3 \\ -6.5 & 15 \end{bmatrix}$ |

[1] Rows are attackers, columns are defenders. Each configured scenario has different transition probabilities and rewards.

## 4. Experiments

In this section, we formulated attack–defense scenarios in IoDM based on smart home IoT, industrial IoT, and medical IoT, respectively. Along with the comparison of the efficiencies of attack and defensive deception according to the vulnerabilities of IoT devices

and system nodes in the configured scenarios, we conducted sensitivity analyses according to the main parameters by related actors and compared the analysis results.

### 4.1. Configuration of Organizational IoT-Enabled Scenarios and Detailed Simulation Parameters

To experiment the efficiency of defensive deception, all complex experimental metrics, attack–defense flows, and topologies related to IoT organizational network operation based on unique IoT sensors and systems, IoT switches, unified threat management (UTM) devices, and firewall solutions are calculated. In this case, each scenario is classified according to the IoT device or sub-farm network structure dedicated according to the unique organizational operation purposes such as industrial type and medical type. Furthermore, each scenario is also composed in detail with the spatiotemporal performances and costs by IoT actors, intrusion–defense distinction criteria and final objectives, CKC, attack and deceptive defense sequences, and other assumptions such as CVE and CVSS. In addition, variables such as surface and contact information by IoT actors with a limited field of view, occupation rate competition related to intrusion success and defense success by episode, and feedback from attack and deception signal execution are also subdivided as shown in Tables A4 and A5 of Appendix A and applied differently by scenario.

Accordingly, the determined smart home IoT, industrial IoT, and medical IoT-based organizational operation scenarios and the unique general sum attack–defense competition interfaces related to individual scenarios have the following premises for performance:

- IoT attacker's action standard: An attacker intrudes an IoT-based organizational network from the outside and carries out attacks on the internal IoT sensor device, system, security solution, and sub-farm network. The final intrusion target points are selected as single or multiple IoT nodes, and the intrusion continues according to the CKC stages until the state of equilibrium is reached or the permitted attack time expires. As the episode unfolds, the intrusion target is changed according to the collected defender's vulnerable contact points, attack surface information, and the effectiveness of the discount factor-based intelligence, and an attack chain is formed toward the production of the highest microscopic reward by time point

- IoT attacker's APT attack strategy: The attacker considers finally occupying the target point defined in the current episode as the top priority strategy, and fans out all possible attacks based on the attack graph and attack tree defined outside before the initial intrusion to optimize the attack gains. In cases where the rate of success in achieving the first priority goal is lower than the possessed threshold, the attack strategy changes to select and occupy the next best attack target. This is done by changing the attack origins by episode or activating the tactics of lateral movement in a different direction of movement. However, if the attacker makes an error in judgment due to changes in the defender's IoT network or deceiving acts, then the actual initiative of the attacker in the competitive relation may disappear as the attacker's initial perception is biased toward defender dominance.

- IoT defender action standard: The defender monitors all IoT device units, security solutions, and sub-farm networks. However, due to the limited resources, the security solutions protecting the affiliated IoT nodes cannot be operated in a timely manner in all cases. The defensive deception signaling strategies by internal IoT sensor devices are not optimized based on the resources or the attenuation ratio of deceptive signals increases. Therefore, in cases where the defense contact point for countermeasures is selected wrongfully, the defense for the corresponding IoT device, farm network, and security solution will fail, and consequently, microscopic defense gains will decrease and the ripple effect of chain attacks in the CKC subsequent stages will increase. The defender's goals are differently configured according to the importance levels by affiliated IoT nodes and the possibility of expression of vulnerabilities. However, they are aimed at proactively avoiding attacks on all IoT nodes and immediately defending against detected attackers for as long as possible. In the case of critical system-based IoT scenarios, the sub-farm networks composed of VLANs by IoT sensor devices

are interconnected, or all periodic snapshots, restorations, and sandboxing isolation actions are carried out to improve safety.

- IoT defender's deception strategy: The defender performs MTD- and decoy-based defensive deceptive actions for all affiliated IoT nodes in the topology based on leading signaling to disrupt and mislead the attacker's judgment. However, since such deception alone cannot completely block intruding IoT attackers, reactive defense is also carried out. Since the defender's reactive response is only possible on the basis of a single IoT device and a sub-farm network, an appropriate prevention point or response point deployment strategy should be produced depending on the available resources.

- Definition of scenario compensation: The concept of compensation that determines actions in the general sum game in IoDM is made into a constant based on CVE and CVSS related to the IoT sensor device. However, it can be dynamically increased or decreased according to the attacker's APT level and present situation of intrusion, the level of defender deception-response, and the state of occupation rates by the actor, etc. so that they become variables

- Calculation of scenario compensation: IoT attackers by episode win rewards when they succeeded in CKC stages, while the defender wins rewards when they succeeded in proactive–reactive defense against the attacker's CKC intrusion act. The reward values are differently defined in relation to the importance levels, correlations, ripple effects, and vulnerabilities of IoT nodes, and microscopic reward values are determined accordingly. When the final simulation has ended, each IoT actor calculates the microscopic reward value.

- Definition of episode discount factor: A concept of discount factors was introduced to force quick decisions by IoT actors and prevent the fixation of the macroscopic equilibrium. That is, attackers and defenders cannot postpone microscopic decisions at the present time unlimitedly for more than a hundred episodes, and the MILP solution space to obtain maximum rewards is also limited by the cut-off.

- Simulation termination condition: With reference to an attacker, this is when the attacker reaches the intrusion target point configured at the present time point, carried out gradual exploitation based on the vulnerable point of contact, and thereafter occupies it based on the action of the object (AoO). With reference to a defender, it is when the defender neutralizes the detected attacker's CKC-based act of intrusion and expels the attacker by completely depleting the resources. After the simulation ends, along with the reward values optimized by episode, the probability values related to the main metrics based on the efforts, costs, and utilities of the attacker and the defender are finally returned.

- Addition of the concept of security solutions dedicated to IoT: UTM is a primary security terminal that combines intrusion detection system (IDS) and intrusion prevention system (IPS) functions. IDS and IPS functions are performed to identify, detect, and block threats from IoT attackers. Then, the firewall is an access control within an IoT-based organizational network and operates as a secondary security terminal that distinguishes the validity of legitimate users and authorizes them. Depending on the scenarios, these security solutions are diversified horizontally and become double-modular-redundant or triple-modular-redundant vertically. In this case, the potential vulnerability is only assumed as a hole-based vulnerable contact point for external IoT attackers to bypass, and unlike the affiliated IoT nodes, the attacker's act of occupation through remote code execution, privilege escalation, etc. are limited or not performed.

Based on the premises for performance above, the IoT organization scenario related to each IoT domain is established as follows.

The focus points related to the IoT operation strategies such as the configured IoT sensor device and system, network topology structure, and security solution are also defined in detail differently by scenario.

- Scenario 1 (Figure 4): 'Smart home IoT-Based open organizational network topology'.

(1) IoT node configuration: IEEE 802.11 WiFi device ('Wireless access point'), smart TV ('Smart furniture'), wall pad ('Wall pad'), smart door lock ('Smart door lock').

(2) IoT security solution configuration: Single IoT-enabled UTM device with some user access control and authorization functions.

(3) IoT attacker's intrusion goal: The best goal is to obtain firmware authority for the smart door lock through contamination of input factors of an unauthenticated external IoT attacker, and then achieve improper unlocking. When it seems impossible to acquire the authority for the target smart door lock, the target is changed into a WiFi device that has more vulnerable contact points and a wide attack surface to achieve suboptimal intrusion, and then differentially carry out chain attack actions of pivoting with a wall pad, etc. in parallel.

(4) IoT defender's defense goal: While performing MTD and decoy-based proactive evasion and deception for all IoT nodes and security solutions, depleting all utilities related to the validity of the defender intelligence possessed by a professional IoT attacker and an attempt to enter the CKC stage to completely achieve reactive blocking of remote intrusion within the IoT network.

(5) Attacker's major intrusion path and fanout tactic: Outside the IoT network → attempt to intrude inside → bypass IoT-enabled UTM's access control and detection policy → carry out an immediate search for and access to a smart door lock with a vulnerability in privilege elevation due to parameter contamination. When continuing access or attack seems impossible, progressively carry out differential intrusion, pivoting, occupation, and search by IoT node according to other vulnerabilities → finally achieve the intrusion into and occupation of the smart door lock.

(6) Defender's major defense sequences and normalization strategy: Perform MTD based on the shuffling of the entire organizational networks by IoT and host layer information. Decoying based on dynamic sandboxing is also performed in parallel. Configure proactive deception process → carry out UTM-based real-time monitoring according to limited resources → detect the intruding attacker and control the access of the attacker, and carry out reactive blocking → force the attackers to make inferior judgments and maximize the required costs → achieve complete expulsion to the outside.

- Scenario 2 (Figure 5): 'Closed organizational network topology based on the industrial IoT applied with a dual modular redundancy (DMR) security solution dedicated to IoT'.

   (1) IoT node configuration: multiple industrial sensor camera device-based farm networks ('Sensor camera device farm'), multiple industrial thermometer-based farm networks ('Thermometer device farm'), multiple industrial meter-based farm networks ('Meter device farm').

   (2) IoT security solution configuration: Two UTMs and two firewalls horizontally interconnected and vertically DMRed.

   (3) IoT attacker's intrusion goal: The best goal is to obtain power meter firmware authority through remote code execution based on stack-based buffer overflow of the web application, and then to achieve the securing of a second intrusion vector for other interconnected industrial IoT devices. When it seems impossible to obtain the authority for a single target meter or the relevant farm network, achieve the sub-optimal intrusion by damaging the thermometer that can cause very large human-property losses or damaging the confidentiality of protected resources based on visual images and changing the target into an industrial sensor camera device from which meaningful real-time information can be taken over. Pivoting is carried out in parallel in the affiliated sub-farm network from the initially occupied node as a starting point.

   (4) IoT defender's defense goal: Same as that in Scenario 1

(5)    Attacker's major intrusion path and fanout tactic attacker: Outside of IoT network → attempt to intrude → bypass the policy of DMRed IoT-enabled UTM and firewall → carry out an immediate search for and access to the power meter that has vulnerabilities in the execution of remote codes and privilege escalation due to buffer overflow. When continuing the search and the access or attack seem impossible, carry out pivoting to another meter within the same sub-farm network or change the intrusion route to another farm network → achieve the intrusion into the power meter and final occupation.

(6)    Defender's major defense sequences and normalization strategy: Force defender dominant deceiving, disturbing, and isolating actions with MTD based on shuffling by each of the entire IoT nodes and decoying based on multitenancy sandbox → carry out reinforced monitoring and detection of multiplexed UTM and firewall → carry out reactive blocking and expulsion of the IoT attacker that intruded any sub-farm network in parallel → achieve complete expulsion from the upper network.

- Scenario 3 (Figure 6): 'Closed organizational network topology based on Medical IoT applied with a triple modular redundant (TMR) security solution dedicated to IoT'.

(1)    IoT node configuration: farm network including CT device and duplexed virtual clone device ('CT device farm'), farm network including MRI device and duplexed virtual clone device ('MRI device farm'), farm network including in the fusion pump, electrosurgical unit-based medical sensor device, and, respectively, duplexed virtual clone devices ('Sensor device farm').

(2)    IoT security solution configuration: Vertically TMRed integrated IoT-enabled UTMs and three firewalls.

(3)    IoT attacker's intrusion goal: The best goal is to obtain the sensitive data of a medical infusion pump applied with inappropriate privilege handling routines, seize the administrator privilege, and achieve the possibility of a direct threat to human life through lateral movement to similar IoT devices within the same farm network. When it seems impossible to gain control over a single-target infusion pump or electrosurgical unit or the relevant farm network, change the target into CT and MRI devices that can threaten the lives of many patients with a greater scope of malfunction while taking over the patients' sensitive personal information thereby achieving suboptimal intrusion. Pivoting is carried out in parallel in the affiliated sub-farm network from the initially occupied node as a starting point.

(4)    IoT defender's defense goal: Same as that in Scenario 2.

(5)    Attacker's major intrusion route and fanout tactics: Outside the IoT network → attempt to intrude inside → bypass the policy of TMRed IoT-enabled UTM and firewall → carry out an immediate search for and access to a medical injection pump sensor device with vulnerability in privilege escalation due to sensitive data disclosure. When it seems impossible to continue the attack, carry out pivoting to another medical sensor device in the same sub-farm network or change the intrusion route into another farm network → carry out intrusion into the drug infusion pump and neutralization and destruction of the backed up image → achieve the final occupation.

(6)    Defender's major defense sequences and normalization strategy: Perform MTD and Decoy-based proactive deception process → carry out triplication-based additional monitoring while ensuring snapshot-based IoT node image integrity in parallel → Detect IoT attacker intrusion, reactively block and expel the attacker, and carry out restoration to normal states with random snapshots → achieve complete expulsion to the outside

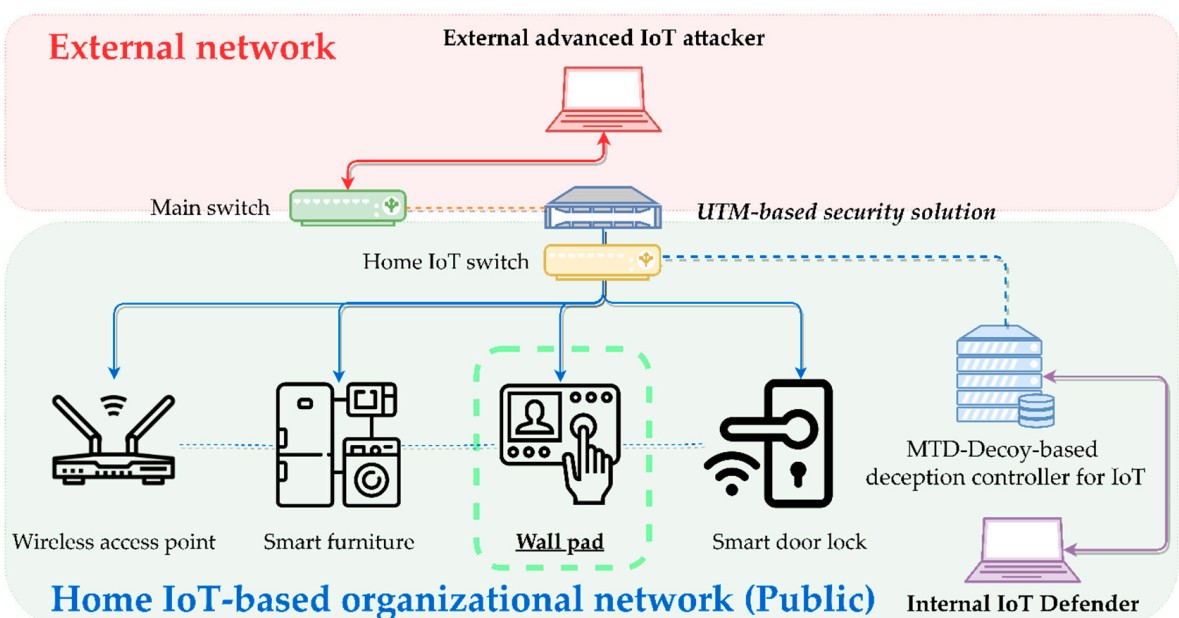

**Figure 4.** Overview of the home IoT-enabled topology with an organizational attack–defense strategy for Scenario 1.

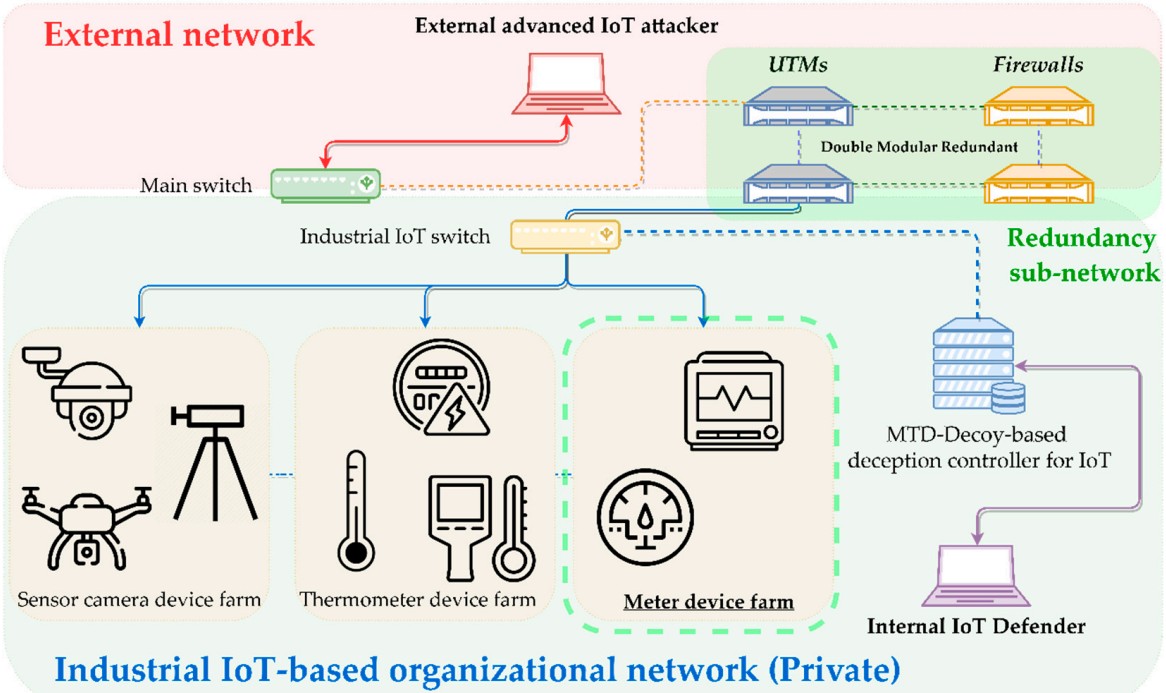

**Figure 5.** Overview of the industrial IoT-enabled topology with an organizational attack–defense strategy for Scenario 2.

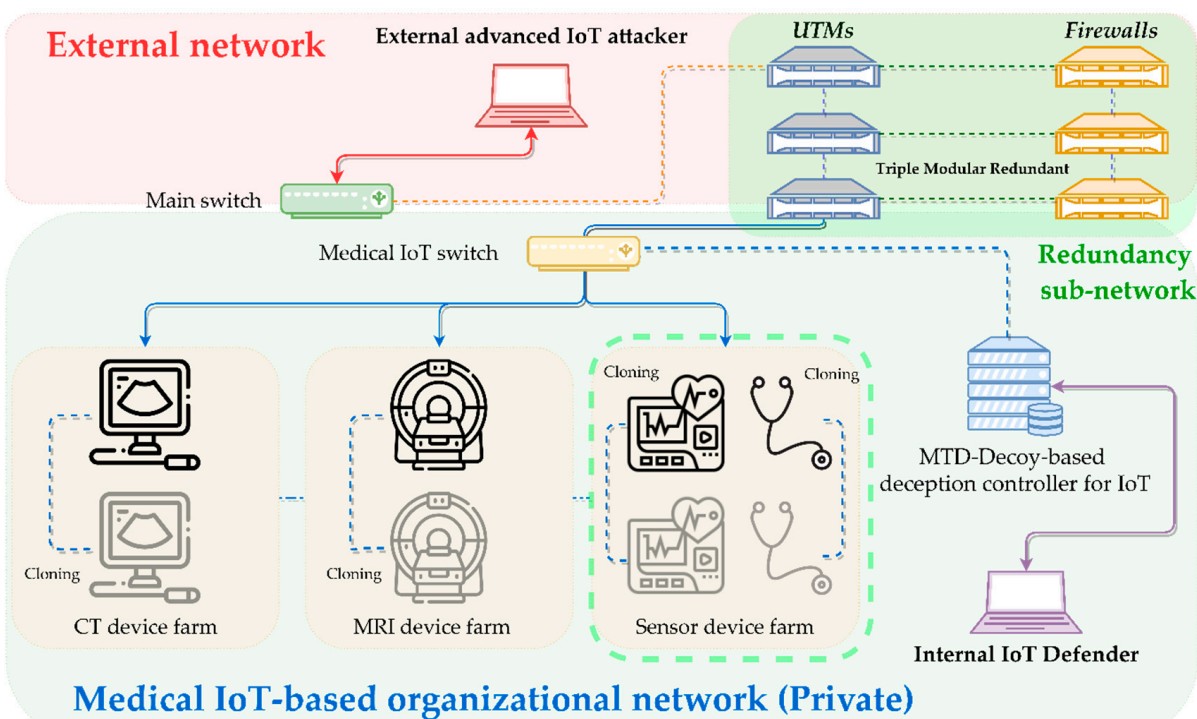

**Figure 6.** Overview of the medical IoT-enabled topology with an organizational attack–defense strategy for Scenario 3.

To classify by IoT scenario, configure the major MITER CVE vulnerabilities and CVSS scores as shown in Table A3 of Appendix A. The formulated IoT vulnerabilities were selected from the list of public CVEs that produce high exploitability scores in relation to smart homes, industrial control systems, SCADA, and large medical organization environments. This is because they can be easily applied as standard indexes for addition and deduction of compensation during general sum-based signaling and feedback based on intelligence, visibility, and occupation rate information possessed by each actor.

In addition, not only the attack graph and attack tree based on the calculated CVE-based IoT vulnerable contact points, but also the defender's defense sequences are multiplexed as IoT-enabled organizational OSINT-based adaptive mutations as with the deception sequences shown in Figure 7. In this case, direct mutation of unique authorized network information used for internal IoT service supply as a target of direct mutation by IoT-enabled MTD is not suitable for all the service availabilities, channel migration schemes, and prescribed encryption processes unique to the organization. Therefore, virtual communication channel information is established within the range of the network address pool determined in advance at random mutation time intervals among internal IoT nodes or certain authorized remote users. Then, the communication channels are connected to the real network addresses and pair tables held by the relevant IoT nodes to reconfigure the IoT routing tables. To entice attackers to a sandbox-type isolation space despite it having a fingerprint similar to that of the protected real IoT node, the rule table is formulated to have a pair structure with the actual network address possessed by the IoT-enabled node using the virtual communication channel information used by the previous IoT node. In addition, the concept of the judgment of whether to perform dynamic shifting of virtual network information in relation to the changes in major entropies in the relevant IoT network when an arbitrary IoT node is occupied following the attacker's success in partial intrusion is also applied.

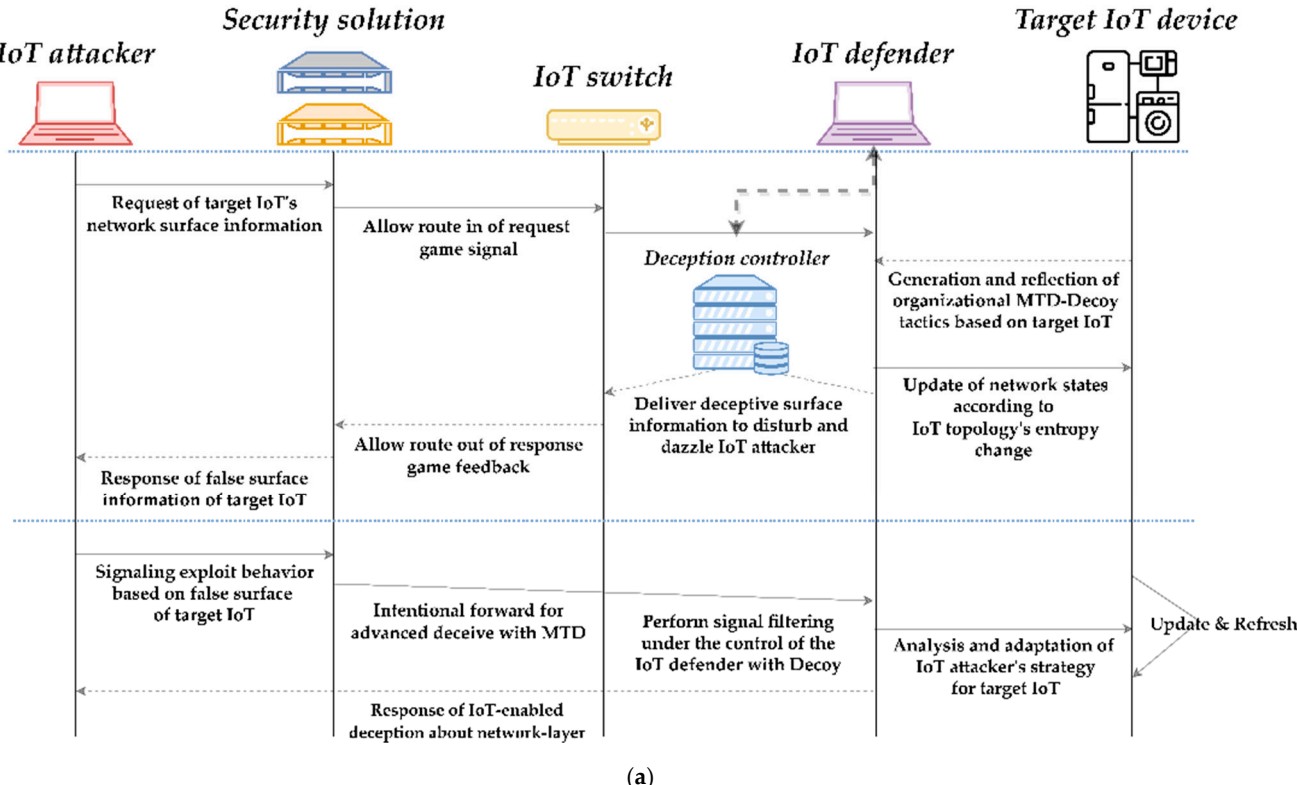

(**a**)

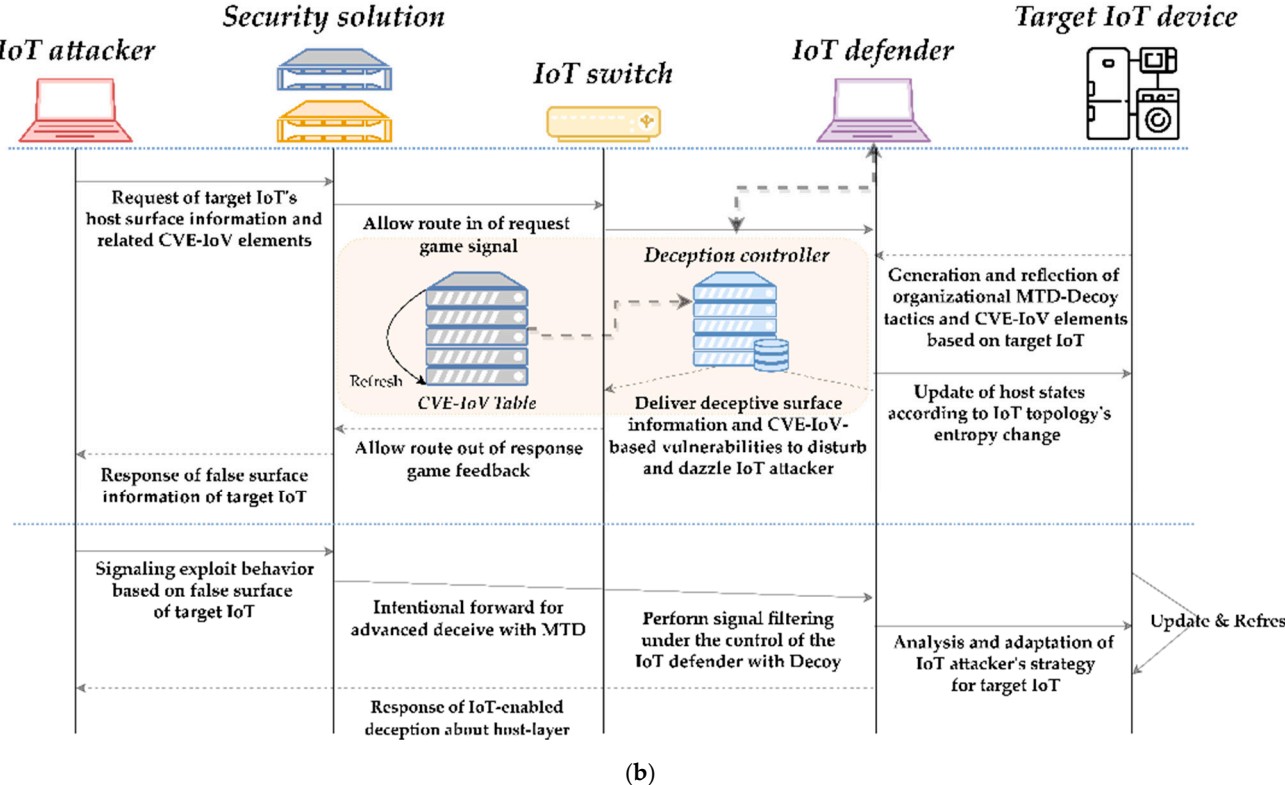

(**b**)

**Figure 7.** Examples of defender's episode-based deceptive sequences with MTD and decoy concepts in IoDM. (**a**) Deceptive sequence beyond the network layer. (**b**) Deceptive sequence below the host layer.

Figure 7a is constructed based on the production of the concept. It is the MTD and decoy-based deceptive signaling sequence when the connection with the internal IoT node selected as a target based on the defender's virtual communication channel information previously collected by an illegal IoT attacker was requested. In other words, when the network communication specifications configured in the request packet by the attacker at the present time point are not allowed within the IoT-based organization network, the relevant attacker is misinformed with false IoT surface size, vulnerability information, and intelligence based on signaling to induce the attacker's cognitive bias. In addition, to artificially deceive an attacker to be easily induced, to enter, or to be isolated in the decoy, virtual communication channel information similar to the network communication specifications requested by the attacker or previously owned by the real IoT node is allocated to an internal decoy or mutated in a direction to minimize the attacker's suspicion, then inserted in the response packet and delivered to the attacker. The attacker whose cognition was disturbed and lured due to their inferior deceptive signaling gradually continues to act toward the maximization of the defender's gains in the relevant episode and completely loses the possibility to reach the final intrusion target point selected in advance. Figure 7b also shows a similar deceptive signaling sequence. However, instead of attempting attacks with network layer information based on IP and socket ports, the attacker carried out attacks in units of IoT-enabled service protocols such as physical and data link layer protocols such as WiFi and Zigbee, network layer protocols such as IPv4 and 6LoWPAN, and application layer protocols such as CoAP and MQTT.

*4.2. Results*

In this section, general sum simulation experiment results related to IoT-enabled MTD and decoy-based defensive deception concepts are produced by IoT-based organizational scenarios defined in IoDM. We also carry out comparison and analysis between decision schemes by major metrics such as the efficiency of attack–defense, costs, and utilities related to the competing actions of attackers and defenders. Each formulated IoT scenario and attack–defense sequence is characterized based on quantified vulnerability scores as shown in Table A3 of Appendix A, along with a POMDP-based state-transition probability matrix as shown in Tables 2, A1 and A2. The overall parameters related to general sum-based deception experiments were also established based on Tables A4 and A5 of Appendix A.

4.2.1. Comparative Analysis of Each IoT-Based Scenario

Figures 8–10 show the result sets normalized by $S_0$, $S_1$, $S_2$, $S_3$ of the POMDP in Figure 3 in relation to the mixed scheme newly configured as the IoT-enabled MTD and decoy concepts customized based on the existing deception and distribution policies [53,54,61,64] were combined with PBNE, BSSG, and partial signaling for the degree of defense success according to gradual changes in the final rewards and discount factors of the defender by the IoT scenario.

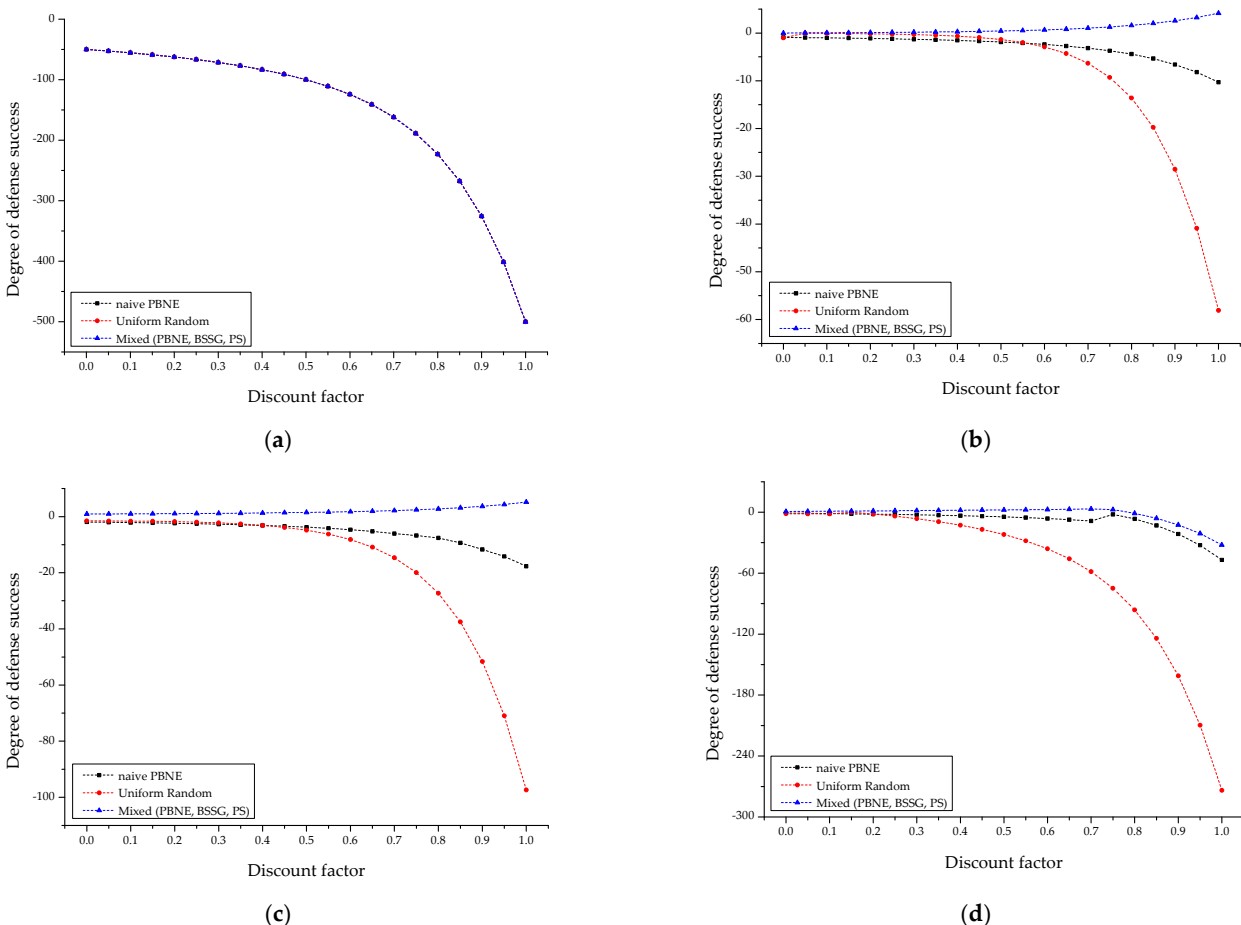

**Figure 8.** Comparison of the attack success probability in each state with Scenario 1. (**a**) $S_0$, (**b**) $S_1$, (**c**) $S_2$, and (**d**) $S_3$.

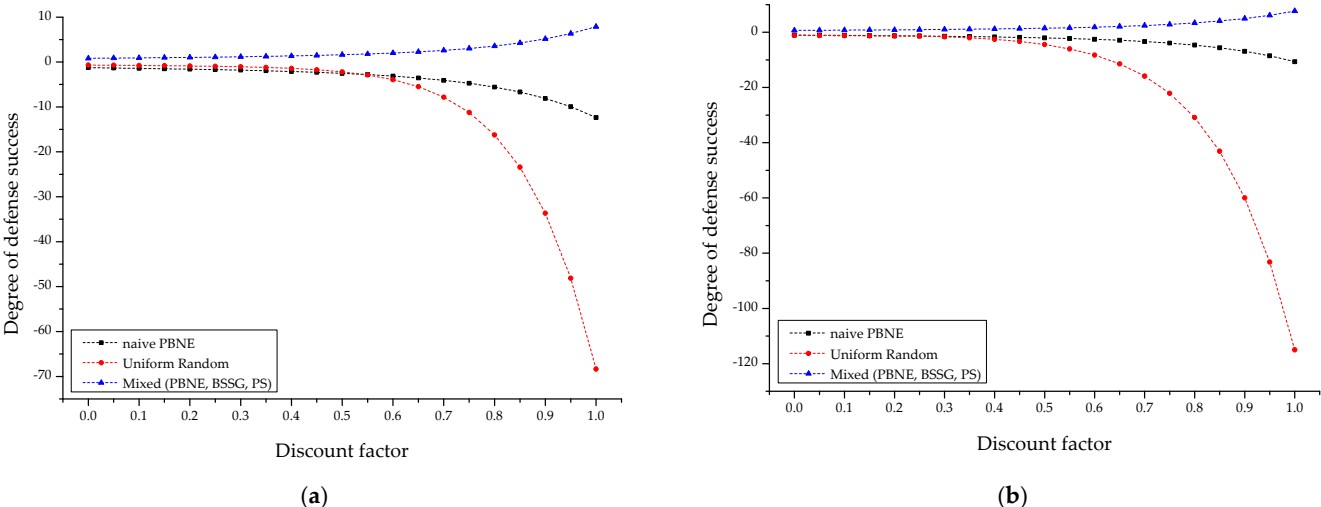

**Figure 9.** *Cont*.

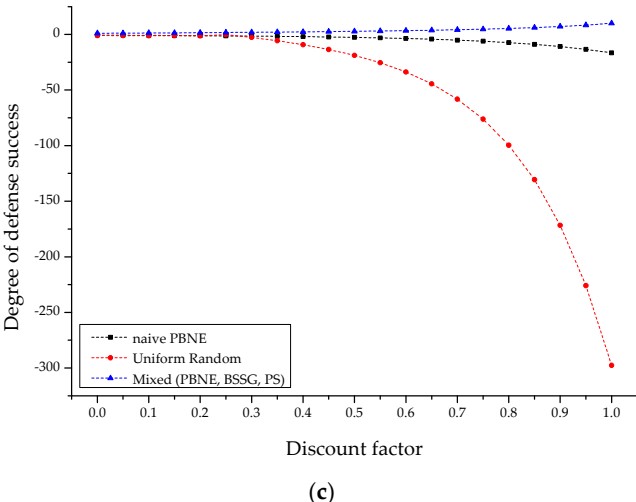

(**c**)

**Figure 9.** Comparison of the attack success probability in each state with Scenario 2. (**a**) $S_1$, (**b**) $S_2$, and (**c**) $S_3$.

(**a**)

(**b**)

(**c**)

**Figure 10.** Comparison of the attack success probability in each state with Scenario 3. (**a**) $S_1$, (**b**) $S_2$, and (**c**) $S_3$.

Figure 8a shows the comparative results calculating the levels of efficiency of success of defense by the defender in scenario 1 according to the discount rate to cut off the

maximum ranges of judgment by actor, thereby forcing quick judgment and increasing the rate of acquisition of microscopic rewards based on $S_0$. When the attacker occupies the final intrusion target selected, or the attacker or the defender expels the attacker, the sum of the defender's total reward decreases exponentially as the discount factor increases. Moreover, the decreased total reward is the same regardless of the applied game decisions strategies related to reward optimization and entering the equilibrium state. That is, when the discount factor is less than 0.7, the compensation has gradients for linear decreases. For example, when the discount factor is 0.3, 0.5, and 0.7, the reward becomes −71.4, −99.9, and −161.9, respectively. However, when the discount factor is 0.7 or higher, the reward decreases exponentially, for example, when the discount factor is 0.8 and 0.9, the reward is −223.1 and −325.6, respectively, which are all extreme compensations. This gradient change pattern is equally derived from $S_0$ in Scenario 2 and Scenario 3. Regardless of the unique characteristics and domain differentiation of the defined IoT scenario, the attacker's spatial and temporal analysis determines the optimal fanout penetration path within the condition that the general sum competition is terminated due to the attacker completely succeeding or failing to capture the target. The signaling of IoT-enabled MTD is calculated so that the validity of the effective attack surface expires while actively attenuating the superiority in the presence of the defender. It is instantaneous and the MTD mutation time interval is shorter. The satellite also stems from the fact that it is enforceable. The gradient pattern change is derived identically from $S_0$ in Scenarios 2 and 3. Regardless of the unique characteristics of the defined IoT scenario and domain distinction, this tendency is attributable to the fact that under the condition where the general sum competition is terminated because the attacker completely succeeded or failed in occupying the target when the judgment of mutation of signaling of IoT-enabled MTD calculated so that the attacker's spatiotemporal dominance determines that the optimal intrusion route is actively attenuated in the presence of the defender and the validity of the valid attack surface is expired is immediate and the time interval of MTD mutation is short, the attacker's gain can be suppressed and asymmetric subordinance can be enforced. In addition to $S_0$ in the determined terminated state, the pattern of the gradient drop is identical in Figure 8b–d. In a situation where the success or failure of an attack in the smart home IoT network is finally determined, to improve the proactive efficiency of IoT-enabled MTD and decoy deception strategies' bias toward the attacker's initial cognitive judgment toward the dominance of the defender with a single parameter, the discount factor should be controlled to not exceed 0.7 from the viewpoint of the IoT attacker. This is done by attenuating the spatiotemporal size of the defender's intelligence that can be judged or by making false deceptive information into disinformation.

Figure 8b–d shows the comparative results that calculated the efficiency of success of defense by the defender in Scenario 1 based on $S_1$, $S_2$, and $S_3$, respectively. In Figure 8b, $S_1$ is related to the primary defense of a single IoT-enabled UTM that fostered user access control and authorization functions. Even when the discount factor increases so that the adaptability of the defensive MTD and decoy is reduced, relatively high asymmetric dominance is given to the attacker. The reduction of the efficiency of success of defense of decision strategies, excluding the Uniform Random decision strategy, is not large and the discount factor converges between −6.5 and 2.5 instead of 0.9. In addition, in the case of the mixed decision strategy that combined PBNE, BSSG, and partial signaling, the efficiency of success of defense rather increases as the discount factor increases, unlike other decision strategies. In this case, the mixed decision strategy derives an average improvement in defense efficiency ranging from 28% to 139% compared to the PBNE decision strategy, which is the baseline. This tendency is attributable to the fact that the efficiency of the MTD and decoy deception process is improved for the mixed decision strategy. The estimation of the probability distribution is related to the leader–follower-based mutual feedback, signaling-based cognitive perturbating, disinformation, attacker's intrusion preference, observable-determinable present situations of activity by IoT node, and the degree of changes in attacker entropy, which does not exist in other decision

strategies. The above tendencies are also attributable to the fact that the mixed decision strategy can derive additional improvement in the performance because it adds more subdivided deceptive elements to the IP and socket port-based network layer, protocol, and service-based host layer while securing practicality as an IoT scenario based on major organizational domains. Furthermore, this tendency is attributable to the fact that the Uniform Random decision strategy, which determines a single candidate node, which is the target of the defender's proactive deceptive signaling and monitoring, reactive detection and blocking, and expulsion with a uniform probability distribution, unconditionally distributes invasion probability uniformly to the upper entity that is unique and transfers high ripple effect vertically as with UTM in Scenario 1 even when no external attack is carried out so that extremely low performance is derived because it cannot be judged that the attacker can carry out intrusion in earnest only after breaking through the relevant UTM. These aspects are similar in Scenario 2 in Figure 9 and Scenario 3 in Figure 10, i.e., to minimize the decrease in the efficiency of the deceptive defense against a single UTM in the smart home IoT, select the mixed decision strategy regardless of the value of the discount factor to ensure defender dominance.

In Figure 8c, where an attacker who bypassed the upper-level security solution intrudes the IoT-based organization network to perform an initial search and attempts exploitation for the first time, the Uniform Random decision strategy shows the lowest defense success efficiency and derives reward values lower than those in $S_1$. In this case, the mixed decision strategy derives defense efficiency ranging from 23% to 139% compared to the naive PBNE decision strategy. This tendency is attributable to the fact that the wall pad, which is the attacker's final intrusion target point, has Wi-Fi as a repeater and exerts a high ripple effect with a single vulnerability, and the performance efficiency of the mixed decision strategy with mixed detailed decision strategies is high. This aspect is also the same in Scenario 2 in Figure 9 and Scenario 3 in Figure 10, i.e., to minimize both the internal reconnaissance and the possibility of success of initial exploitation of an IoT attacker who is searching for potentially vulnerable contact points to improve attack effectiveness after intruding the IoT-based organizational network for the first time, select the mixed decision strategy regardless of the value of the discount factor to disturb the attacker in the general sum-based competitive environment.

In Figure 8d, $S_3$ is related to the achievement of the final intrusion goal following CKC-based lateral movement and the advancement of the successive chain after the success of the initial exploitation. The mixed decision strategy derives a defense success efficiency ranging from 7% to 52% to the PBNE decision strategy, thereby showing a sharply reduced performance, unlike before. This tendency, which is different from Figure 8a–c, is attributable to the fact that since the IoT attacker found out that there is only a single real vulnerability of the wall pad selected as the final intrusion target point by them, they completely identified the relevant vulnerability as the last hole into which all remaining resources possessed by them can be intensively assigned. To reduce and actively block the possibility of success of the final intrusion by the IoT, which performs lateral movements and many rough privilege escalation attacks after succeeding in the initial occupation of a certain IoT node within an IoT-based organization network, select the mixed decision strategy while quickly emitting deceptive signals so that the discount factor is formed between 0.7 and 0.75. This will make noises so that the attacker's cognitive perturbation is configured toward the dominance of the defender.

Figures 9 and 10 show the sets of discount factor-based results for critical system scenarios where UTM and firewall devices are vertically DMRed or TMRed and the major affiliated IoT nodes grouped are configured in-depth with sub-farm structures by role. Here, all of the single ripple effects and additional effects are distributed in the industrial IoT-based closed network and the medical IoT-based closed network, respectively. The dominance relations of the defense efficiencies by the presented decision strategy technique are the same as those in Figure 8.

Figure 9a shows the defensive success efficiency of the defender in Scenario 2 based on $S_1$. The defensive success efficiencies of decision strategies other than the mixed decision strategy rapidly decrease when the discount factor increased to 0.75 to reach $-10$ and $-50$, respectively, and converge when the discount factor became 0.95. On the contrary, the mixed decision strategy yielded defense success efficiencies improved by 68% through 172% compared to the PBNE decision strategy. This tendency is attributable to the fact that the UTM solution in the previous Scenario 1 was interconnected with the firewalls and diversified horizontally and multiplexed vertically so that the transferred ripple effects concentrated by the sub-industrial IoT node and farm network were dispersed. It is also attributable to the fact that since the attacker's final intrusion target point was changed from a single IoT node to a farm network composed of multiple IoT power meters, and the intrusion routes were limited due to the strict closed network policy of ICS, the transferred ripple effects statically produced by affiliated IoT node were probabilistically dispersed further. To minimize the reduction of the efficiency of defensive deception by the multiplexed UTM and firewall solutions, select the mixed decision strategy regardless of the value of the discount factor to continue proactive deceptive actions against external attackers who wish to bypass while securing real occupation rates by the IoT node. This will ultimately improve the security of the industrial IoT-based organizational network based on redundancy and diversity. Figure 9b shows the defensive success efficiency of defenders in Scenario 2 based on $S_2$. The mixed decision strategy yields improved defense success efficiency by 51% through 158% compared to the naive PBNE decision strategy. To actively reduce the possibility of success of attacks by an IoT attacker attempting initial intrusion and initial exploitation of a farm network in which multiple IoT power meters are grouped, the mixed decision strategy can be also adopted regardless of the value of the discount factor. Figure 9c shows the deriving of the defense success efficiency in Scenario 2 based on $S_3$. Selecting the mixed decision strategy, deceptive counter-measures against the IoT attacker who wishes to occupy the final intrusion target point can be formulated. The mixed decision strategy is shown to produce defense success efficiencies improved by 33% through 107% compared to the PBNE decision strategy.

Finally, the results of experiments of Scenario 3 related to the TMRed UTM, firewall solutions, and the operation strategy that considered restorability and resilience by cloning the candidate IoT node images based on snapshots are compared and analyzed with the result sets shown in Figure 10. Figure 10a shows the defense efficiency based on $S_1$ against external IoT attackers who should bypass the TMRed IoT-enabled security solutions. The defensive success efficiencies of decision strategies other than the mixed decision strategy rapidly decreased from when the discount factor increased to 0.75 to reach $-22$ and $-50$, respectively, and converge when the discount factor became 0.95. Conversely, the mixed decision strategy is shown to yield defense efficiencies improved by 79% through 212% compared to the PBNE decision strategy. Since sharp positive gradients were formed when the discount factor increased to 0.7, the defense efficiencies were finally derived to be between $+8$ and $+10$. This tendency is attributable to the fact that in the TMRed security operation environment in Scenario 3, the defender's risk due to external IoT attackers' CKC-based intrusion acts is lower compared to that in other scenarios with different security operation environments. On the other hand, the attacker's consumption of costs and utilities for reconnaissance, search, and exploit attempts can be forced and is also attributable to the fact that the concept of cyber resilience based on redundancy and the concept of cyber agility based on diversity can be included in the upper layer of the relevant IoT network. Figure 10b shows the results of analysis of the efficiency of defense efficiency based on $S_2$ against attackers who attempt initial exploitation after reconnaissance of the lower farm network that possesses cloned IoT node snapshot images. The mixed decision strategy yielded defense efficiencies improved by 54% through 168% compared to the PBNE decision strategy so that the defense efficiency increased from when the discount factor reached 0.75 and finally converged between $+7$ and $+11$. This tendency is attributable to the fact that even if the restorability and resilience levels of

the IoT node are improved with redundancy-based additional snapshot images, if the attacker's initial occupation is carried out immediately after their exploitation within the time length for the snapshot configuration, the IoT nodes will be reverted to the state similar the defender intelligence possessed by the attacker already at the relevant time point so that the attacker's spatiotemporal dominance in the attack surface will be still preserved at a high level. Figure 10c shows the analysis of the efficiency of defense based on $S_3$ against the IoT attacker who wants to occupy the final intrusion target point, indicating that the mixed decision strategy yielded defense efficiencies improved by 37% through 133% compared to the naive PBNE decision strategy so that the defense efficiency increased with positive gradients finally converging between +17 and +21.

For the defender to respond dominantly to all of the unequal reward value-adding acts of external IoT attackers within the industrial IoT and medical IoT-based organizational networks in which the redundancy and diversity-based security were finally maximized, the deceptive signaling should select the mixed decision strategy regardless of the value of the discount factor.

### 4.2.2. Sensitivity Analysis with Decision Strategy

To subdivide the concepts of microscopic–macroscopic rewards that can be acquired through general sum-based attack–defense competition in the IoDM to calculate, compare, and analyze the performances and costs of major metrics possessed by individual actors, sensitivity analyses are performed in detail for each of Uniform Random, simple PBNE, and mixed decision strategies. Figure 11 shows the sets of results normalized based on industrial IoT-based Scenario 2 to calculate the efficiency of deceptive defense against availability attacks, integrity attacks, and defender intelligence-based in-depth exploitation attacks by external IoT attackers. Figure 12 shows the sets of the results of reclassification of detailed metrics by IoT-enabled MTD and decoy-based major deception parameter and normalization of the results thereafter based on industrial IoT-based Scenario 2 in order to derive the defense costs and resources consumed according to the major factor values of MTD and decoy.

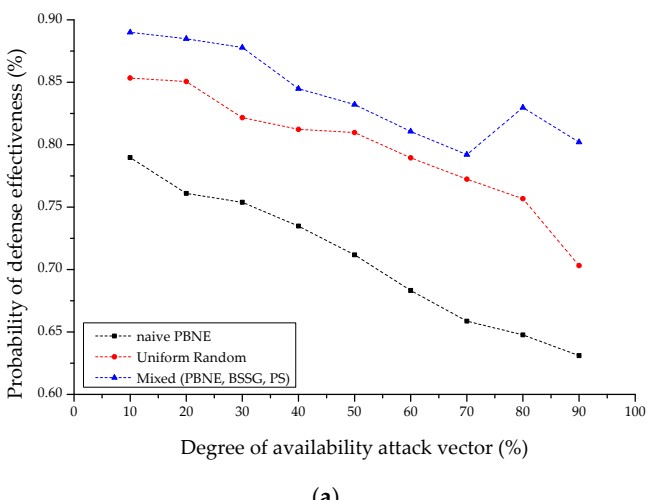

(**a**)

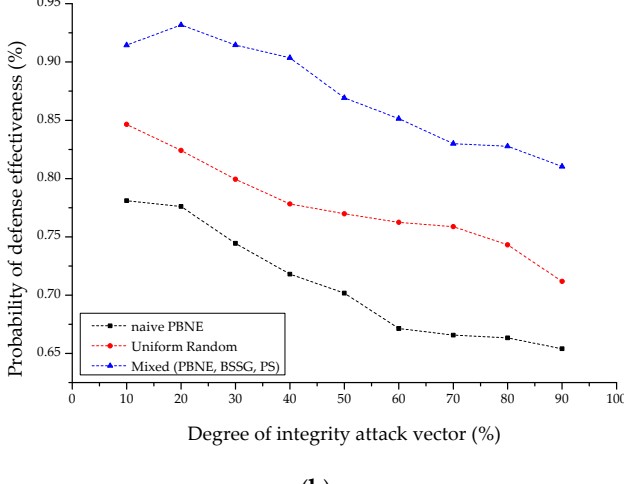

(**b**)

**Figure 11.** *Cont.*

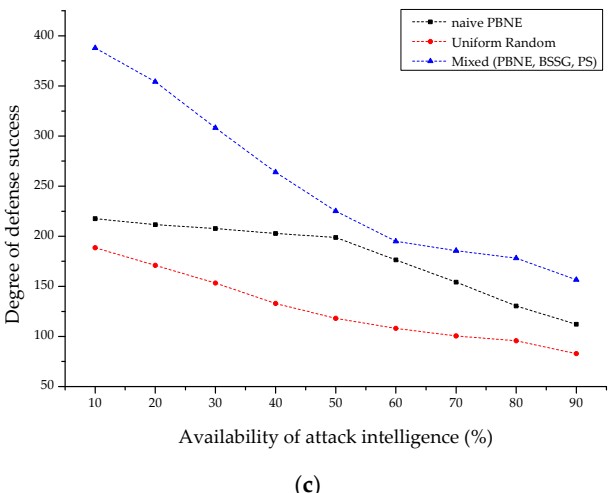

(**c**)

**Figure 11.** Comparison of the defense effectiveness each decision scheme in Scenario2. (**a**) Availability attack. (**b**) Integrity attack. (**c**) Validity of intelligence.

(**a**)

(**b**)

(**c**)

**Figure 12.** Comparison of the defense utility with deceptive signal each decision scheme in Scenario2. (**a**) artificially exposed vulnerability level, (**b**) mutation time slot length, and (**c**) number of IoT-enabled decoy hole.

Figure 11a shows the results of comparison of the levels of availability attacks that interrupt packet transmission/reception of internal IoT nodes and rapidly consume resources allocated to communication sockets attacker dominantly among the intrusion vectors of the external IoT attackers who have intruded the industrial IoT network based on packet delivery ratios (PDR) excluding the environmental losses. The mixed decision strategy yields higher deceptive defense efficiencies against availability attacks by at least 14.8% on average compared to the simple PBNE decision strategy and by at least 7.2% on average compared to the Uniform Random strategy. Figure 11b shows the results of a comparison of the levels of integrity attacks that damage the packets exchanged by internal IoT nodes or sharply deteriorate the quality of service related to the IoT communication protocol based on PDR and packet error rate (PER). The mixed decision strategy yields higher deceptive defense efficiencies against integrity attacks by at least 16.2% on average compared to the naive PBNE decision strategy and by at least 8.3% on average compared to the Uniform Random strategy. Figure 11c shows the results of comparison and analysis of the levels of the validity of defender intelligence utilized by attackers when they perform exploitation and occupation of a certain IoT node corresponding to the current intrusion target point. Similarly, the mixed decision strategy yields higher deceptive defense efficiencies against attack surface-based exploitation by at least 38% on average compared to the naive PBNE decision strategy and by at least 82% on average compared to the Uniform Random strategy.

Figure 12a shows the results of comparison and analysis of the levels of vulnerability of the industrial IoT that carries out artificial disinformation among various deceptive techniques applied when an internal IoT defender performs deceptive signaling based on IoT-enabled MTD and decoy to disturb and partialize external IoT attackers based on the defense costs and resources consumed. Based on the same deception performance, the mixed decision strategy non-uniformly derives defense overhead at a slightly lower rate of 4.2% on average compared to the Uniform Random decision strategy but increased by at least 8.4% on average compared to the naive PBNE decision strategy. Figure 12b shows the result of comparison and analysis of the IoT-enabled MTD applied to shuffle the fingerprints of affiliated IoT nodes as a proactive evasion irrespective of the identification and detection of IoT attackers based on the defense costs and resources consumed according to the mutation time slot length of the IoT-enabled MTD. Based on the same MTD performance, the mixed decision strategy derives the defense utility at a slightly lower rate of 3.4% on average compared to the Uniform Random decision strategy but increased by at least 7.9 times on average compared to the naive PBNE decision strategy. Finally, Figure 12c shows the results of comparison and analysis of the IoT-enabled decoy holes distributed to extremely reduce the efficiency of the successive attack chains by reactively inducing the intruding IoT attacker and isolating the attacker in the sandbox environment based on the defense costs and resources consumed according to the number of distributed IoT-enabled decoy holes. Based on the same decoy performance, the mixed decision strategy derives a defense utility at a reduced by at least by 5.8 times on average compared to the Uniform Random decision strategy but increased by at least 3.8 times on average compared to the simple PBNE decision strategy.

In this case, this tendency expressed in all the corresponding result sets is attributable to the fact that the mixed decision strategy has a lower or similar deceptive overhead compared to the Uniform Random decision strategy. The Uniform Random decision strategy has a uniform random probability distribution but derives larger defense costs than the naive PBNE decision strategy, which is a single decision strategy. In addition, it is attributable to the fact that since IoT-enabled deception tactics such as disinformation-based signaling, artificial disclosure, deceptive perturbation, virtualized IoT specification information, and cognitive disturbance scheme according to the decoy path are also added separately, the minimum required defender utilities by each episode are also increasing. In summary, this proves that the mixed decision strategy derives significantly improved deceptive defense efficiency compared to other decision strategies. Furthermore, the defense efforts required by the mixed decision strategy are a little lower or almost similar

to those required by the Uniform Random decision strategy, although higher than those required by the simple PBNE decision strategy.

## 5. Discussion

In this study, we proposed IoDM applied with PBNE and BSSG, a partial signaling-based general sum game, and POMDP to quantify macroscopic decision-making strategies for real-time attack–defense competition in the IoT domain. In addition, we formulated a proactive IoT-enabled MTD that defender dominantly optimizes mutation time slot, shuffling policy, mutation set, mutation sampling, etc., and a reactive IoT-enabled decoy based on unique organizational IoT components. Macroscopic deception strategies optimized by IoT scenarios could be produced based on the general sum game equilibrium, and microscopic IoT deception tactics related to the decoy performance of MTD and decoys could also be derived as results subdivided by major metrics. Furthermore, an approach that minimizes the use of limited resources in the IoT network and performance degradation and maximizes security could also be abstracted at the scenario level. Among major differentiated experimental results, the mixed decision strategy derived performances improved by at least 70% on average compared to the Uniform Random and simple PBNE decision strategy. In sensitivity analysis for the efficiency of deceptive defense to ensure CIA in industrial IoT-based organizational topology, deceptive performance improved by at least 20% on average compared to other decision schemes. Regarding overheads by MTD and decoy-based variables, it was also proved through comparative analysis that load balancing more optimized for the IoT operating environment can be performed if the improved deceptive performance is considered.

However, threats to the validity of this study and potential improvement measures are:

- The issue of demand for strategization due to the limited range of decisions: In an actual IoT system and network environment, attack–defense actors subjectively process asymmetric information available at the present time and decide on actions after making incomplete judgments. This proves that a simple naive game strategy that models dynamic decision-making under the premise that actors in an uncertain situation have a consistent view is not generalizable. This study tried to mitigate the problem by conceptualizing disinformation-based partial signaling game tactics and adaptively changing the state-transition probability according to changes in the attack–defense state in the modeling related to conflicts of interest between the IoT attack–defense actors; however, potential side effects will still remain. Accordingly, the hyper game theory [41], which is an unbalanced metagame, should be quantitatively utilized to formulate the inferior decision-making flow when the actors in competitive relationships by episode are 'induced' to select the best strategy due to subjective or false beliefs, differences in information and view, misperception, and perceived uncertain judgment, etc. In addition, various solution spaces for the relevant modeling of conflicts and methods to produce optimal solutions configured based on the uniqueness of the IoT domain (e.g., frequent additional interventions by unspecified internal and external actors, impossibility to cut off decision strategies due to explosive increases in interconnected channels, etc.) should be processed. Furthermore, approaches such as FlipIt games [67–69] and other various game theories [70] should be considered.
- Substantiality of IoDM and optimization problem: The IoT organizational characteristics intellectualized in the game in this IoDM were quantified as based conditions for sequential entries into the game equilibrium state through the calculation of the state-transition probability in POMDP and reward values after preprocessing the CVEs or related IoVs disclosed. However, they will be different from the unique policies or beliefs related to the actual organization that operates the IoT device-system-network and will show clear practical differences from the network-separated IoT organization environment according to the range of disclosure of vulnerable information and the IoT domain classification. In addition, since the defined IoT-based CVEs and IoVs were simply limited to CVSS scoring-based parameterization rather than considering

the correlation or ripple effect between IoT device-system-network specifications, additional augmentation and normalization should be also considered.

- The scalability problem of the IoT deception model: Since this study carried out experiments to reduce the attack-exploration surface through defense actors' performance of IoT-enabled MTD and decoy-based deceptive actions and performance optimization within the scope of scenarios configured in the IoT domain, we did not consider scenarios or other domains other than the ones described. In addition, since the IoT specifications emulated in the game are also formulated as attack–defense probability values based on random scenario configurations, the probability values should be abstracted at a more stratified organizational IoT domain classification level and the appropriate IoT operation process should be conceptualized.
- IoT-enabled MTD and decoy applicability problem: The main experimental results were derived using MTD metrics based on mutation period and intensity, candidate target, and sampling and decoy metrics based on decoying elements, properties, and paths. However, an improved solution set should be produced by securing a wider solution space. An approach to hierarchically modeling complex correlations between IoT systems is required. Therefore, it is necessary to improve the approach based on reinforcement learning, deep neural network, and graph neural network [71], which have been actively studied recently in the field of deception.

## 6. Conclusions

In this study, we proposed a PBNE, BSSG, partial signaling-based general sum game foreground- and POMDP state-transition background-based organizational model and named it IoT-based organizational deception modeling (IoDM). In addition, the concepts of proactive MTD and reactive decoys were also formulated to foster defender-dominant deception strategies and detailed parameters based on the characteristics of the IoT device–system network. We also performed simulations based on all IoT scenarios, sequences, and CVE and IoV tables dedicated to the main domains such as smart home, industry, and medical care. Through this, by sequentially modeling the competitive decision-making process between CVE-IoV-based attackers and MTD-decoy-based deceptive defense actors, it was possible to calculate deception efficiency optimized for each major IoT domain and organizational IoT environment. In addition, it was possible to simulate the temporal and spatial superiority of each IoT actor when the general sum-based equilibrium state is reached according to the gradual progress of game episodes. As a quantitative comparison result, it was possible to significantly increase the deceptive defense efficiency of the IoT defender by more than 70% on average, and also to improve the deceptive performance of the CIA standard by more than 20%, and the optimized defender cost was also demonstrated by key metrics and parameters. In the future, to improve the reliability of the deception efficiency of the proposed IoDM and to conceptualize IoT-based threat intelligence, we plan to conduct advanced follow-up studies to diversify deceptive countermeasures and anti-deception tactics composed of unique IoT attack vectors. In addition, we also intend to expand and improve the decision solution space of this study by additionally adopting the deceptive hypergame theory.

**Author Contributions:** Conceptualization, S.S.; methodology, S.S.; software, S.S.; validation, S.S. and D.K.; formal analysis, S.S. and D.K.; investigation, S.S.; resources, S.S.; data curation, S.S.; writing—original draft preparation, S.S. and D.K.; writing—review and editing, S.S. and D.K.; visualization, S.S.; supervision, D.K.; project administration, S.S. and D.K; funding acquisition, D.K. All authors have read and agreed to the published version of the manuscript.

**Funding:** This research received no external funding.

**Data Availability Statement:** Not applicable.

**Acknowledgments:** This work was supported by a Kyonggi University Research Grant (2021).

**Conflicts of Interest:** The authors declare no conflict of interest.

# Appendix A

**Table A1.** Probability matrix of transition and semi-constant reward value with payoff strategy in IoT-enabled Scenario 2.

| State | Probability of Transition in Scenario 2 | Reward Value for Defender |
|---|---|---|
| $S_0$ | $[(1, 0, 0, 0)]$ | $[\,-60\,]$ |
| $S_1$ | $\begin{bmatrix} (0, 1, 0, 0) & (0, 1, 0, 0) & (0, 1, 0, 0) \\ (0, 0.25, 0.75, 0) & (0, 0.925, 0.075, 0) & (0, 0.65, 0.35, 0) \\ (0, 0.25, 0.75, 0) & (0, 0.68, 0.32, 0) & (0, 0.91, 0.09, 0) \end{bmatrix}$ | $\begin{bmatrix} 0 & -2 & -2 \\ -15 & 15 & -2 \\ -12.9 & -2 & 12.9 \end{bmatrix}$ |
| $S_2$ | $\begin{bmatrix} (0, 0, 1, 0) & (0, 0, 1, 0) & (0, 0, 1, 0) & (0, 0, 1, 0) \\ (0, 0, 0.2, 0.8) & (0, 0.75, 0.25, 0) & (0, 0.2, 0.8, 0) & (0, 0.2, 0.8, 0) \\ (0, 0, 0.25, 0.75) & (0, 0.3, 0.7, 0) & (0, 0.75, 0.25, 0) & (0, 0.2, 0.8, 0) \\ (0, 0, 0.2, 0.8) & (0, 0.3, 0.7, 0) & (0, 0.2, 0.8, 0) & (0, 0.75, 0.25, 0) \end{bmatrix}$ | $\begin{bmatrix} 0 & -1 & -2 & -2 \\ -10 & 9.5 & -2 & -2 \\ -6.5 & -1 & 5.5 & -2 \\ -10 & -1 & -2 & 9.5 \end{bmatrix}$ |
| $S_3$ | $\begin{bmatrix} (0, 0, 0, 1) & (0, 0, 0, 1) \\ (0.75, 0, 0, 0.25) & (0.05, 0.15, 0.6, 0.2) \end{bmatrix}$ | $\begin{bmatrix} 0 & -2 \\ -15 & 31 \end{bmatrix}$ |

**Table A2.** Probability matrix of transition and semi-constant reward value with payoff strategy in IoT-enabled Scenario 3.

| State | Probability of Transition in Scenario 3 | Reward Value for Defender |
|---|---|---|
| $S_0$ | $[(1, 0, 0, 0)]$ | $[\,-55\,]$ |
| $S_1$ | $\begin{bmatrix} (0, 1, 0, 0) & (0, 1, 0, 0) & (0, 1, 0, 0) \\ (0, 0.4, 0.6, 0) & (0, 0.999, 0.001, 0) & (0, 0.75, 0.25, 0) \\ (0, 0.4, 0.6, 0) & (0, 0.72, 0.28, 0) & (0, 0.975, 0.025, 0) \end{bmatrix}$ | $\begin{bmatrix} 0 & -3.5 & -3.5 \\ -17.25 & 17.25 & -3.5 \\ -28 & -3.5 & 28 \end{bmatrix}$ |
| $S_2$ | $\begin{bmatrix} (0, 0, 1, 0) & (0, 0, 1, 0) & (0, 0, 1, 0) & (0, 0, 1, 0) \\ (0, 0, 0.23, 0.77) & (0, 0.8, 0.2, 0) & (0, 0.2, 0.8, 0) & (0, 0.2, 0.8, 0) \\ (0, 0, 0.2, 0.8) & (0, 0.3, 0.7, 0) & (0, 0.8, 0.2, 0) & (0, 0.2, 0.8, 0) \\ (0, 0, 0.2, 0.8) & (0, 0.3, 0.7, 0) & (0, 0.2, 0.8, 0) & (0, 0.8, 0.2, 0) \end{bmatrix}$ | $\begin{bmatrix} 0 & -2 & -3 & -3 \\ -6.95 & 6.95 & -3 & -3 \\ -10.0 & -2 & 10.0 & -3 \\ -10.0 & -2 & -3 & 10.0 \end{bmatrix}$ |
| $S_3$ | $\begin{bmatrix} (0, 0, 0, 1) & (0, 0, 0, 1) \\ (0.65, 0, 0, 0.35) & (0.03, 0.2, 0.55, 0.22) \end{bmatrix}$ | $\begin{bmatrix} 0 & -3.5 \\ -14 & 33 \end{bmatrix}$ |

**Table A3.** CVE and CVSS-based vulnerability table based on IoT devices.

| Scenario | CVE ID | Vulnerability and Related Weakness [1] | Related Node | Exploitability Score in CVSS 2.0 |
|---|---|---|---|---|
| Home IoT-based organizational network (Scenario 1) | CVE-2018-3953 | OS command injection | Wi-Fi | 8.0 |
| | CVE-2021-34991 | Remote code execution with overflow | Wi-Fi | 6.5 |
| | CVE-2021-44632 | Remote code execution with overflow | Wi-Fi | 10.0 |
| | CVE-2020-9759 | Privilege escalation | TV | 10.0 |
| | CVE-2018-4082 | Remote code execution with overflow, Memory corruption, Denial-of-service | TV | 8.6 |
| | CVE-2021-30780 | Privilege escalation | TV | 8.6 |
| | CVE-2019-19163 | Remote code execution | Wall pad | 6.5 |
| | CVE-2019-13143 | Privilege escalation with parameter pollution | Door lock | 10.0 |
| | CVE-2019-12944 | Missing authorization | Door lock | 8.6 |
| | CVE-2020-25223 | Remote code execution | UTM | 10.0 |
| | CVE-2020-17352 | OS command injection | Firewall | 8.0 |
| Industrial IoT-based organizational network (Scenario 2) | CVE-2016-9155 | Improper access control | Sensor camera | 10.0 |
| | CVE-2018-10661 | Bypass of restriction | Sensor camera | 10.0 |
| | CVE-2017-3209 | Incorrect default permissions and missing authentication | Sensor camera | 6.5 |
| | CVE-2019-3944 | Incorrect default permissions with deauthentication | Sensor camera | 10.0 |
| | CVE-2013-4860 | Incorrect default permissions | Thermometer | 6.5 |
| | CVE-2018-11315 | Bypass of restriction and related permissions | Thermometer | 6.5 |
| | CVE-2017-9944 | Improper privilege management | Meter device | 10.0 |
| | CVE-2021-44165 | Remote code execution with overflow | Meter device | 10.0 |
| | CVE-2018-0101 | Remote code execution with double-free | UTM | 10.0 |
| | CVE-2021-34787 | Bypass of restriction | Firewall | 8.6 |

**Table A3.** *Cont.*

| Scenario | CVE ID | Vulnerability and Related Weakness [1] | Related Node | Exploitability Score in CVSS 2.0 |
|---|---|---|---|---|
| Medical IoT-based organizational network (Scenario 3) | CVE-2018-8857 | Hard-coded credentials | CT device | 3.9 |
| | CVE-2020-25175 | Weak encryption of protected credentials | CT device | 10.0 |
| | CVE-2021-26262 | Improper access control | MRI device | 10.0 |
| | CVE-2020-25179 | Obtain credentials | MRI device | 10.0 |
| | CVE-2019-13543 | Hard-coded credentials | Medical sensor | 10.0 |
| | CVE-2020-12041 | Incorrect permission assignment | Medical sensor | 10.0 |
| | CVE-2020-25165 | Improper authentication | Medical sensor | 10.0 |
| | CVE-2021-36807 | Remote code execution with SQL injection | UTM | 8.0 |
| | CVE-2022-1040 | Remote code execution with bypass | Firewall | 10.0 |

[1] These vulnerabilities are CVSS-based numerical constant and used when optimizing attack–defense rewards in IoDM.

**Table A4.** The major design parameters, their meanings, and the representative values in IoDM-based testbed.

| MTD Parameter | Value | Decoy Parameter | Value |
|---|---|---|---|
| Time slot length for periodic mutation (s) | 1–86,400 | Activation time (s) | 0–518,000 |
| Mutation batch pool size | 32–2048 | Number of decoying hosts | 1–5 |
| Bellman-based mutation sampling size | 8–2048 | Number of decoying services | 1–10 |
| Mutation shuffling tactic [61] | Random, GeneticHeuristic | Number of decoying vulnerabilities | 1–10 |
| Mutation period decision scheme [61] | Random, Adaptive, Hybrid | Number of decoying beacons | 1–20 |
| Number of false surface views in topology | $0-2^4$ | Number of decoying signals | 1–10 |
| Maximum number of branches in attack graph-tree | $2^3$ | Level of attack severity with vulnerability | L-M-H |
| Maximum number of deceptive signaling | $0-2^5$ | Maximum number of compromised decoys | 0–5 |
| Mutation range of security solutions | $0-2^4$ | Maximum number of sandboxing holes | 0–1 |
| Mutation range of IPv4 addresses | $2^8-2^{36}$ | Feedback rate for adaptation each episode (%) | $10^{-4}-10^{-2}$ |
| Mutation range of port numbers | $2^{10}-2^{16}$ | Probability of interaction each attack step (%) | 0–100 |
| Mutation range of OS fingerprints | $0-2^4$ | Probability of cloning (%) | 1–90 |
| Mutation range of protocol services | $0-2^4$ | Probability of mimicking (%) | 1–80 |
| Mutation range of vulnerabilities | $2^1-2^4$ | Probability of enticingness in decoy (%) | 30–100 |
| Probability of disinformation (%) | 1–100 | Probability of conspicuousness in decoy (%) | 50–100 |
| Probability of artificial disclosure (%) | 1–100 | Probability of variability in decoy (%) | 10–100 |
| Probability of reliability of deceptive signal (%) | 1–100 | Probability of differentiability in decoy (%) | 70–100 |

**Table A5.** The major decision strategy arguments with MTD and decoy in IoDM-based testbed.

| Parameter | Value |
|---|---|
| Simulation time (s) | 7200–518,000 |
| Number of simulation run for Monte Calro | 100 |
| Attack time (s) | 3600–518,000 |
| Defense time (s) | 3600–518,000 |
| Number of scenarios | 3 (Home IoT, Industrial IoT, Medical IoT) |
| Number of real IoT devices each node | 1 |
| Switch mode | Virtual local area network (VLAN) |
| Number of CKC phases with IoT attacker | 4–7 |
| Validity of attak surface (%) | 10–90 |
| Efficiency of packet drop attacks (%) | 10–90 |
| Efficiency of packet modify attacks (%) | 10–90 |
| Discount factor (%) | 0–100 |
| Number of vulnerability and weakness each node | 1–4 |
| Methodology of constructing attack paths | Attack graph, attack tree |
| Operating system | Windows 10 |
| Language | Python 3.7.5 (Anaconda) |
| MILP-based general-sum game solver | Gurobi optimizer 9.0 |

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
