# Peer review of "IoDM: A Study on a IoT-Based Organizational Deception Modeling with Adaptive General-Sum Game Competition"

_electronics, doi:10.3390/electronics11101623_

Round 1
Reviewer 1 Report
The article is well written and easy to understand. However, few of my feedback can be considered to improve the quality of the paper but all are not necessary.
- Introduction may be improved, adding the highlights and the problem statements.
- You could improve writing, link better the ideas flow in the Introduction.
- Review references because some of them are unstandardized.
- The conclusion needs improvements towards major claimed contribution.
- Write some future directions in the conclusion section.
- The difference between your proposal and related works is not clear, you could to details better. I suggest add a comparative table in ''Related Literature'' to contrast your solution in front of related works.
- You could discuss the relationship between your solution and past literature. You can cite following papers: https://onlinelibrary.wiley.com/doi/abs/10.1002/ett.4329 , https://link.springer.com/article/10.1007/s10586-020-03046-w
Author Response
Dear reviewer,
Thank you, your comments on our paper “IoDM: A Study on a IoT-based Organizational Deception Modeling with Adaptive General-Sum Game Competition”.
I would like to resubmit a revised version of our prior manuscript (Manuscript ID: electronics-1719544).
We have incorporated all comments and suggestions from you as best we can in the revised version of our manuscript. This letter includes both the reviewer comments and our response to these comments.
In our responses, we have explained the changes made to the manuscript based on the suggestion of the reviewers. We have also mentioned the location of these revisions, to make it easier for the reviewers to examine these changes. Wherever possible, we have provided an excerpt of the revised text.
Yours sincerely,
The Authors.

Reviewer 2 Report
This paper is well written. However, I would suggest 2 things.
1) The introduction can be improved by adding some more background work.
2) Once the introduction is improved, then a separate section should be added which presents the comparison of the technique proposed with the state-of-the-art existing methods.
Author Response

(The authors gave the same response as above.)

Reviewer 3 Report
- Paper is well written. Authors should add a little background of the study and limitations of the existing works and clearly explain the contributions at the end of the introduction.
- Some Paragraphs in the paper can be merged and some long paragraphs can be split into two.
- i see, figures needs to be improved. this is a must required.
- Equations are not cited and well defined, it is recommended to authors to defines the variables and explain.
- The first paragraph of introduction can cite this paper:
https://www.mdpi.com/1424-8220/22/6/2087 - Include limitations of the study and future work in the conclusion section.
Author Response

(The authors gave the same response as above.)

Round 2
Reviewer 1 Report
Authors updated the paper and no further update requires from my side.